# DATA-AWARE TRAINING QUALITY MONITORING AND CERTIFICATION FOR RELIABLE DEEP LEARNING

## ABSTRACT

Deep learning models excel at capturing complex representations through sequential layers of linear and non-linear transformations, yet their inherent black-box nature and multi-modal training landscape raise critical concerns about reliability, robustness, and safety, particularly in high-stakes applications. To address these challenges, we introduce YES training bounds, a novel framework for real-time, data-aware certification and monitoring of neural network training. The YES bounds evaluate the efficiency of data utilization and optimization dynamics, providing an effective tool for assessing progress and detecting suboptimal behavior during training. Our experiments show that the YES bounds offer insights beyond conventional local optimization perspectives, such as identifying when training losses plateau in suboptimal regions. Validated on both synthetic and real data, including image denoising tasks, the bounds prove effective in certifying training quality and guiding adjustments to enhance model performance. By integrating these bounds into a color-coded cloud-based monitoring system, we offer a powerful tool for real-time evaluation, setting a new standard for training quality assurance in deep learning.

## 1 INTRODUCTION

Deep learning models have become crucial for tackling complex computational problems, owing to the rich representations they develop through their multi-layered structures and non-linear transformations LeCun et al. (2015); Goodfellow et al. (2016). Despite their remarkable effectiveness, these models are often perceived as black boxes, raising concerns related to their robustness, reliability, and safety. As neural networks become increasingly integral to critical applications, ensuring that they are properly trained and perform as intended is paramount.

To evaluate the training performance and a network's ability to store a model after training (i.e., achieve zero loss), one approach is to statistically analyze neural networks under certain assumptions. This has been done for networks with thresholding activation functions like ReLU, where researchers have determined the number of parameters needed to achieve full memory capacity Vershynin (2020). It is well-known that for ReLU-based neural networks (NNs), once a sufficient number of weights is reached, the network can achieve full memory capacity or even zero loss in some cases. In Oymak & Soltanolkotabi (2019), the authors theoretically demonstrate that in the over-parameterization regime, the stochastic gradient descent (SGD) algorithm can converge to the global minimum. However, these methods are statistical in nature and rely on specific assumptions about the input data and the model, which may limit their applicability.

The most widely encountered scenario in mathematical optimization, including for training deep neural networks, is that the true optimum value of the objective function is not known *a priori*— i.e., it is unclear to what extent the minimization objective can be reduced. Therefore, finding a way to evaluate the effectiveness of optimizers in minimizing the training objectives is an intriguing pursuit. In this paper, we introduce a novel framework for *data-aware certification and monitoring* of deep neural network training, which we refer to as the *YES training bounds*. These bounds aim to provide a qualified answer to the question: *Is the neural network being properly trained by the data and the optimizer (YES or NO)?* The certificates are *data-aware* in the sense that they take into account the specific structure and properties of the training data, allowing for more precise and tailored bounds on the training performance. This means that rather than relying on generic or

overly conservative estimates, the guarantees reflect the actual dataset being used, providing more relevant insights into the training process. They are also *real-time*, fulfilling a long-standing desire in the deep learning community for tools that can evaluate and certify the progress made in training a neural network as it unfolds. By offering concrete, data-specific evaluations during training, these guarantees allow practitioners to confidently assess when sufficient learning has taken place and whether the model is converging toward optimal training, all without the need for extensive post-hoc analysis or probabilistic approximations.

To facilitate the practical application of the YES bounds, we offer an intuitive *cloud-based monitoring system* that visually represents the training progress. This system uses a color-coded scheme—**red** (ineffective training), **yellow** (caution, training non-optimal), and **green** (effective training in progress)—to certify the training status in real time; see Fig. 2. As the field of artificial intelligence (AI) continues to evolve, there is a growing emphasis on *AI safety* and *regulation* to ensure that AI systems meet the highest standards of reliability and effectiveness Howar & Hungar; Tran et al. (2023); Aird—Affiliate (2023); Bengio et al. (2024); Balagurunathan et al. (2021); Brundage et al. (2020); Hendrycks & Mazeika (2022); Marques-Silva & Ignatiev (2022); Qi et al. (2024); Pinto et al. (2015); Zhong et al. (2023); Wing (2021); Li et al. (2023). The proposed YES training bounds and the associated training cloud system present a promising pathway toward establishing a benchmark for the AI industry, regulators, and users alike. We envision a future where these bounds are widely adopted, serving as a standardized measure to evaluate and ensure the optimal training performance of AI systems across diverse applications. This standardization could play a crucial role in fostering trust and accountability within the AI ecosystem.

The YES bounds and their associated cloud system primarily serve as a *sanity check for the optimizer*—the mechanism driving the training process. This system continuously challenges the optimizer with intelligently crafted examples grounded in mathematically rigorous heuristics and tailored to the layer-wise architecture of neural networks. By doing so, the bounds lay the proper groundwork for standardization of training practices in deep learning. Two key observations support this assertion:

1. *The Bounds Go Beyond Local Optimization Perspectives:* Traditional optimization approaches in neural networks predominantly focus on local information—examining the immediate vicinity of the optimization landscape, such as attraction domains and local optima. However, numerical examples demonstrate that training losses can often plateau in regions where the YES bounds and the associated cloud unequivocally indicate suboptimality. This phenomenon underscores the bounds' ability to transcend local optimization insights, providing a more global perspective on training efficacy.

2. *Deterministic Certification Without Randomization:* Unlike methods that rely on randomization around the current training state to certify local or neighborhood optimality, the YES bounds operate deterministically. They do not produce varying certification results across different training realizations, even when initialized identically or following similar optimization paths. This determinism is particularly advantageous for standardization, as it ensures consistent and reproducible determinations of training quality. By eliminating the variability introduced by randomization, the YES bounds present a robust and reliable candidate for establishing standardized training benchmarks in deep learning.

The rest of the paper is organized as follows: In Section 2, we present the preliminaries of our work and outline the architecture of the model to which we apply the YES bounds. Section 3 draws attention to a fundamental bound for single-layer NNs that will later aid in the development of the YES bounds. In Section 4, we extend this idea to deep neural networks, accounting for multiple layers and the nonlinearity of activation functions. Section 5 details our numerical analysis, evaluating the training performance on both synthetic and real data across various applications, such as phase retrieval and image denoising, demonstrating the capability of the proposed bound in assessing AI reliability.

*Notation:* Throughout this paper, we use bold lowercase and bold uppercase letters for vectors and matrices, respectively. We represent a vector $\mathbf{x}$ and a matrix $\mathbf{B}$ in terms of their elements as $\mathbf{x} = [x_i]$ and $\mathbf{B} = [B_{i,j}]$, respectively. $(\cdot)^\top$ is the vector/matrix transpose. The Frobenius norm of a matrix $\mathbf{B} \in \mathbb{C}^{M \times N}$ is defined as $\|\mathbf{B}\|_{\mathrm{F}} = \sqrt{\sum_{r=1}^{M} \sum_{s=1}^{N} |b_{rs}|^2}$, where $b_{rs}$ is the $(r, s)$-th entry

of $\mathbf{B}$. Given a matrix $\mathbf{B}$, we define the operator $[\mathbf{B}]_+$ as $\max\{B_{i,j}, 0\}$. $\mathbf{B}^\dagger$ is the Moore-Penrose pseudoinverse of $\mathbf{B}$.

## 2 PRELIMINARIES

Let $\mathbf{A}_k$ and $\mathbf{b}_k$ denote the weight matrix and bias vector for layer $k \in [K]$ of the network, respectively. Suppose the input and output data are represented by matrices $\mathbf{X} \in \mathbb{R}^{n \times d}$ and $\mathbf{Y} \in \mathbb{R}^{m \times d}$, respectively, such that

$$\mathbf{X} = [\ \mathbf{x}_1\ |\ \cdots\ |\ \mathbf{x}_d\ ], \quad \mathbf{Y} = [\ \mathbf{y}_1\ |\ \cdots\ |\ \mathbf{y}_d\ ], \tag{1}$$

where $\{\mathbf{x}_i \in \mathbb{R}^n\}_{i=1}^d$ and $\{\mathbf{y}_i \in \mathbb{R}^m\}_{i=1}^d$ denote $d$ observations of $n$-dimensional and $m$-dimensional input and output features, respectively. Define the matrices $\{\mathbf{B}_k\}_{k=1}^K$ as $\mathbf{B}_k = [\ \mathbf{b}_k\ |\ \cdots\ |\ \mathbf{b}_k\ ] \in \mathbb{R}^{m \times d}, \quad k \in [K]$. We consider two closely-related training losses for such a DNN employing a nonlinear activation function $\Omega(.)$, namely

$$\mathcal{L}_0\left(\{\mathbf{A}_k\}_{k=1}^K, \mathbf{X}, \mathbf{Y}\right) \triangleq \|\Omega(\mathbf{A}_K\Omega(\mathbf{A}_{K-1}\Omega(\cdots\Omega(\mathbf{A}_1\mathbf{X} + \mathbf{B}_1)\cdots + \mathbf{B}_{K-1}) + \mathbf{B}_K) - \mathbf{Y}\|_F^2, \tag{2}$$

and

$$\mathcal{L}_K\left(\{\mathbf{A}_k\}_{k=1}^K, \{\mathbf{Y}_k\}_{k=2}^K\right) \triangleq \sum_{k=1}^K \|\Omega(\mathbf{A}_k\mathbf{Y}_k + \mathbf{B}_k) - \mathbf{Y}_{k+1}\|_F^2, \tag{3}$$

which is to be minimized with respect to $\{\mathbf{A}_k\}$ and $\{\mathbf{Y}_k\}$ by setting $(\mathbf{Y}_1, \mathbf{Y}_{K+1}) = (\mathbf{X}, \mathbf{Y})$. For our purpose, we will use the following notations:

$$\begin{aligned}\tilde{\mathbf{A}}_k &= [\ \mathbf{A}_k\ |\ \mathbf{b}_k\ ], \\ \tilde{\mathbf{Y}}_k &= [\ \mathbf{Y}_k^\top\ |\ \mathbf{1}\ ]^\top.\end{aligned} \tag{4}$$

Using these notations, the training loss expression in (3) can be reformulated as follows:

$$\mathcal{L}_K\left(\{\tilde{\mathbf{A}}_k\}_{k=1}^K, \{\tilde{\mathbf{Y}}_k\}_{k=2}^K\right) \triangleq \sum_{k=1}^K \|\Omega(\tilde{\mathbf{A}}_k\tilde{\mathbf{Y}}_k) - \mathbf{Y}_{k+1}\|_F^2. \tag{5}$$

Note that any nonlinearity in DNNs can be dealt with by imposing the output structure of the activation function on $\{\mathbf{Y}_k\}$. For instance, for a ReLU activation function, we only need $\mathbf{Y}_k \geq \mathbf{0}$, $k \in \{2, \cdots, K\}$. With this in mind, one may consider the alternative constrained quadratic program:

$$\underset{\{\tilde{\mathbf{A}}_k\}_{k=1}^K,\ \{\tilde{\mathbf{Y}}_k\}_{k=2}^K}{\text{minimize}} \sum_{k=1}^K \|\tilde{\mathbf{A}}_k\tilde{\mathbf{Y}}_k - \mathbf{Y}_{k+1}\|_F^2 \tag{6}$$

$$\text{subject to} \qquad \tilde{\mathbf{Y}}_k \in H_\Omega,$$

where $H_\Omega$ denotes the matrix space created by applying $\Omega$. This representation, which disregards the activation function and imposes a space constraint on the solution, is useful in establishing the foundation for our bounds in subsequent sections. For the rest of the paper, we present our formulations without considering the effects of bias terms. However, these formulations can be easily extended to include bias terms by substituting $\{\mathbf{A}_k\}$ and $\{\mathbf{Y}_k\}$ with $\{\tilde{\mathbf{A}}_k\}$ and $\{\tilde{\mathbf{Y}}_k\}$, respectively.

## 3 AN OPTIMALITY BOUND FOR SINGLE-LAYER NEURAL NETWORKS

We begin by considering a one-layer neural network where the goal is to approximate the function $f : \mathbb{R}^n \to \mathbb{R}_+^m$. Let $\mathbf{A}$ be the weight matrix that we aim to optimize, such that the objective $\|\mathbf{Y} - \Omega(\mathbf{A}\mathbf{X})\|_F^2$ is minimized. Assume $\mathbf{Y}$ is in the feasible set, i.e. the range of a non-linear activation function $\Omega(.)$ of the layer. The weight matrix $\mathbf{A}$ that minimizes the alternative objective $\|\mathbf{Y} - \mathbf{A}\mathbf{X}\|_F^2$ can then be expressed as:

$$\mathbf{A} = \mathbf{Y}\mathbf{X}^\dagger. \tag{7}$$

The minimal achievable loss of training for a one-layer neural network is thus bounded as

$$\|\mathbf{Y} - \Omega(\mathbf{A}^{\text{OPT}}\mathbf{X})\|_F^2 \leq \|\mathbf{Y} - \Omega(\mathbf{Y}\mathbf{X}^\dagger\mathbf{X})\|_F^2, \tag{8}$$

where $\mathbf{A}^{\text{OPT}}$ is the optimal weight matrix that minimizes the objective $\|\mathbf{Y} - \Omega(\mathbf{AX})\|_{\text{F}}^2$. For instance, if $\Omega(.)$ is the ReLU function, then we simply have

$$\|\mathbf{Y} - [\mathbf{A}^{\text{OPT}}\mathbf{X}]_+\|_{\text{F}}^2 \leq \|\mathbf{Y} - [\mathbf{YX}^{\dagger}\mathbf{X}]_+\|_{\text{F}}^2. \tag{9}$$

Since the solution in (7) is feasible (not necessarily optimal, although meaningful) for the optimizer of $\|\mathbf{Y} - \Omega(\mathbf{AX})\|_{\text{F}}^2$, a well-designed training stage is generally expected to satisfy the bound in (8).

## 4 THE TRAINING PERFORMANCE BOUNDS FOR MULTI-LAYER NETWORKS

So, what is the case for *depth* and *nonlinearity*? In scenarios where a single-layer is insufficient, neural learning models employ multiple layers with nonlinear activation functions such as ReLU to progressively refine input mappings (leading to *deep* learning). The necessity for multiple layers can be attributed to the complexity of the mapping task in question. The necessity for nonlinear transformations in multi-layer mappings, however, comes from the fact that multiple consecutive linear mappings are in effect equivalent to a single-layer mapping (with a weight matrix equal to the product of linear mapping matrices).

Inspired by our observation in the single-layer case, in the following we introduce the YES training bounds for multi-layer NNs. These bounds aim to provide a qualified answer to the question as to whether a neural network is being properly trained by the data: YES or NO?

### 4.1 THE YES-0 BOUND

Assuming an initial value $\mathbf{Y}_1 = \mathbf{X}$, the network aims to transform $\mathbf{Y}_1$ through intermediate states $\mathbf{Y}_2, \mathbf{Y}_3, \ldots, \mathbf{Y}_K$, finally achieving $\mathbf{Y}_{K+1} = \mathbf{Y}$. A sensible but sub-optimal approach will be to assume at each layer that we aim to project directly to $\mathbf{Y}$, instead of other useful intermediate points $\{\mathbf{Y}_k\}$. considering our one-layer bound, this no-intermediate approach will be equivalent to setting

$$\mathbf{A}_k = \mathbf{YY}_k^{\dagger}, \quad k \in [K]. \tag{10}$$

In particular, when there are no intermediate mappings we are dealing with an order-0 (referred to as YES-0) bound:

$$\mathcal{L}_0\left(\{\mathbf{A}_k^{\text{OPT}}\}_{k=1}^K, \mathbf{X}, \mathbf{Y}\right) \leq \mathrm{B}_{\text{YES-0}} \triangleq \|\mathbf{Y} - \mathbf{Y}_{K+1}\|_{\text{F}}^2, \tag{11}$$

where

$$\mathbf{Y}_{k+1} = \Omega\left(\mathbf{YY}_k^{\dagger}\mathbf{Y}_k\right), \quad k \in [K]. \tag{12}$$

For instance, we have

$$\mathbf{Y}_{k+1} = \left[\mathbf{YY}_k^{\dagger}\mathbf{Y}_k\right]_+, \quad k \in [K], \tag{13}$$

for the ReLU activation function.

Since the central idea behind the creation of the YES-0 bound, in essence, stems from sequential projections, one may readily expect a decreasing behavior from the bound as the number of layers grows large. This is theoretically and numerically verified in Appendix A.

Obtaining the YES-0 bound, which is evaluated for each layer and projects the input of each layer to the final output, can be viewed as a layer-wise optimization with a linear closed-form operator. The YES-0 bound can serve as an immediate benchmark for the assessment of training in deep learning: One can verify whether the training has a YES-0 bound *certificate*, meaning that they are achieving a training loss lower than YES-0. Otherwise, they can attest that the training is not *proper*.

### 4.2 BEYOND THE YES-0 BOUND

Note that the YES-0 bound is an easily calculated bound that may be used to immediately detect if proper (not necessarily optimal) training has been carried out, in the sense that the network weights have been *meaningfully impacted by the training data*. The answer (YES or NO) will provide immediate relief as to whether training has been meaningful at all. More sophisticated bounds, such as the YES bounds of higher degrees may be used to further assess the quality of training, as discussed in the following.

We begin the construction of the promised bounds by examining whether one can enhance the bounds by considering a more conducive route to the output $\mathbf{Y}$.

### 4.2.1 IS DIRECT PATH THE BEST PATH?

Consider a local optimizer tasked with navigating a highly non-convex loss landscape riddled with numerous local minima. Optimizers like SGD may become trapped in suboptimal regions. Interestingly, allowing the optimizer to momentarily increase the loss can enable it to escape these attraction domains and ultimately reach a better minimum. This fact highlights that sometimes, a non-direct path—including temporary increases in the loss function—can be more effective in minimizing the ultimate objective.

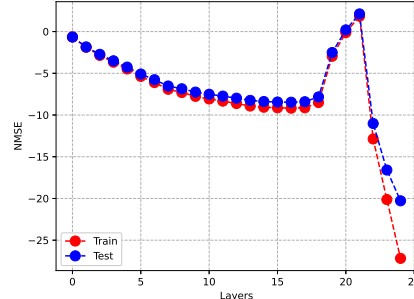

Figure 1: The illustration of a non-monotonic per-layer error (in dB) observed across both training and test stages for a DUN.

This behavior has been widely observed in Deep Unfolding Networks (DUNs) Chen et al. (2022), which are typically formulated based on first-order optimization methods exhibiting monotonic convergence properties. When trained with sufficient data, DUNs learn to behave non-monotonically, allowing temporary increases in loss to escape attraction domains associated with poor local optima. As a result, they can achieve lower overall objectives compared to their original first-order counterparts. Figure 1 illustrates this behavior, showing that the loss at each layer's output does not decrease monotonically.

We will leverage this observation and explore projecting the input from each layer onto a meaningfully selected sequence of intermediate points in lieu of immediately projecting them onto $\mathbf{Y}$. Inspired by the indirect optimization path, this adjustment introduces the notion of the YES bounds of higher degrees, which will be discussed in detail in the following.

### 4.2.2 EXPLORING STRUCTURED TRANSITION THROUGH NONLINEARITY-COMPLIANT SPACES

Enhanced bounds can be established by defining a sequence of useful intermediate points $\{\mathbf{Y}_k\}$ that conform to the nonlinear activation function of the network, i.e. $\mathbf{Y}_k \in H_\Omega$. Let us assume we have such a useful sequence as the outcomes of layers $t_2, \cdots, t_\Sigma$ (where $t_\Sigma = K + 1$) as $\mathbf{Y}_{t_2}^\star, \cdots, \mathbf{Y}_{t_\Sigma}^\star$ (where $\mathbf{Y}_{t_\Sigma}^\star = \mathbf{Y}$). This gives rise to the YES-$\Sigma$ bound, i.e.,

$$\mathcal{L}_0\left(\{\mathbf{A}_k^{\mathrm{OPT}}\}_{k=1}^K, \mathbf{X}, \mathbf{Y}\right) \leq \mathrm{B}_{\text{YES-}\Sigma} \triangleq \|\mathbf{Y} - \mathbf{Y}_{K+1}\|_{\mathrm{F}}^2, \quad \mathbf{Y}_{k+1} = \Omega\left(\mathbf{Y}_{t_\sigma}^\star \mathbf{Y}_k^\dagger \mathbf{Y}_k\right), \quad (14)$$

where $t_{\sigma-1} \leq k \leq t_\sigma, \quad 2 \leq \sigma \leq \Sigma$, with $\mathbf{Y}_1 = \mathbf{X}$. Given a judicious selection of $\mathbf{Y}_{t_2}^\star, \cdots, \mathbf{Y}_{t_\Sigma}^\star$, the latter should provide a tighter error bound compared to the YES-0 bounding approach.

In addition to the local optimization perspective discussed in Sec. 4.2.1, a domain-aware viewpoint is also helpful. While it is fair to say that we hope that by iterative mapping, we get closer and closer to the output of interest, it has also been observed in various machine learning problems that after extensive training (resembling what we can describe as optimal training), the output of some inner layers become something meaningful to domain experts (see, e.g., the literature on object detection in image processing, where the primary task of certain layers is well-understood, such as detecting edges, and certain features). This observation closely associates with the vision put forth above on tightening the YES-0 bound by considering useful and meaningful intermediate mapping points.

The key question in deriving the enhanced YES bounds is thus the judicious designation of intermediate mapping points. One may suggest these two ways:

1. *Problem-specific construction-based sets of intermediate mapping points:* As for the image processing example above, in specific applications, we might have prior knowledge of the structure of the intermediate projection points. This also presents very interesting points of tangency with the literature on model-based deep learning, and in particular, DUNs. In this context, the network layers and the input-ouput relations are strongly connected to a well-established iterative procedure, which can help with identifying suitable intermediate mappings discussed herein.

2. *Training data-driven generation of the mapping points:* This approach takes advantage of the training data to enhance the YES bounds along with the training loss. This is going to be the focus of our proposed YES bounds of *higher degree* in the following.

### 4.2.3   THE YES-$k$ TRAINING BOUNDS ($k \geq 1$)

Leveraging the training data, we utilize the output from a subset of layers acquired during the training process as the intermediate mapping points and incorporate these points into the YES bounds by projecting the input of the layer onto these intermediate outputs rather than relying solely on $\mathbf{Y}$. Specifically, one can utilize $k \in [K-1]$ intermediate mapping points in a $K$ layer network to create associated YES bounds. To better illustrate this idea, we present the example of the higher-degree YES bounds framework for the case of $k = 1$:

1. We set $\mathbf{Y}_1 = \mathbf{X}$.

2. We choose $\mathbf{Y}_2^\star$ as the output of the first layer during the training stage.

3. Following Section 3, we aim to optimize the weight matrix of the first layer $\mathbf{A}_1$, such that the alternative objective $\|\mathbf{Y}_2^\star - \mathbf{A}_1 \mathbf{Y}_1\|_{\mathrm{F}}^2$ is minimized. This is achieved by

$$\mathbf{A}_1 = \mathbf{Y}_2^\star \mathbf{Y}_1^\dagger. \tag{15}$$

4. We then obtain the output of the first layer as $\mathbf{Y}_2 = \Omega\left(\mathbf{Y}_2^\star \mathbf{Y}_1^\dagger \mathbf{Y}_1\right)$.

5. Since we only chose $k = 1$ intermediate point, we then progressively project the input of each layer to the output $\mathbf{Y}$ as

$$\mathbf{Y}_{k+1} = \Omega\left(\mathbf{Y}\mathbf{Y}_k^\dagger \mathbf{Y}_k\right), \quad k \in \{2, \cdots, K\}. \tag{16}$$

6. We compute the resulting error of this process as $\|\mathbf{Y} - \mathbf{Y}_{K+1}\|_{\mathrm{F}}^2$.

7. We repeat steps (3-6) by choosing other intermediate points $\mathbf{Y}_3^\star, \cdots, \mathbf{Y}_K^\star$ in step 2.

8. Take the minimum of all errors obtained in Step 6, leading to the creation of the YES-1 training bound.

The name YES-1 bound takes into account the fact that we have only considered $k = 1$ intermediate point in the above process. Note that the above formulation can be easily extended to $k \in \{2, \cdots, K-1\}$ intermediate points, generating higher degree YES bounds, namely YES-$k$ bounds for $k \in \{2, \cdots, K-1\}$. In contrast to the YES-0 training bound, the YES bounds of higher degree are *real-time*, i.e., they evolve alongside the training loss. It is important to note that YES-$k$ bounds for larger $k$ are not necessarily smaller than those for smaller $k$, particularly at the initial epochs where the training may not suggest excellent intermediate points. Higher degree bounds are, however, highly likely to perform better when the training is in good condition. We numerically validate this phenomenon in Appendix B. In Appendix C, we present a monotonic modification of the YES-$k$ bounds for $k \geq 1$, along with an important observation that increasing the degree, i.e. $k$, does not necessarily improve the YES bounds. In fact, all the bounds remain relatively close to each other. This observation can be beneficial from a computational aspect.

### 4.2.4   THE YES TRAINING CLOUD-SYSTEM FOR QUALITY MONITORING

We now illustrate the integration of the proposed YES bounds with the training process, culminating in the creation of an intuitive training cloud to monitor progress in real time (see Fig. 2). As the YES bounds evolve over epochs, similar to the training loss, they enable users to visually observe the interaction between training performance and the YES bounds. This visualization allows practitioners to track key moments in the training process, such as when the training loss surpasses the YES-0 bound, when it improves beyond the best YES-$k$ bounds, the epochs at which these transitions occur, and how the bounds and training results interact throughout the process.

To help users navigate key transitions during the training process, the YES training cloud system employs a color-coding scheme. A training loss that remains above the YES training cloud—depicted as the red area—indicates ineffective training. When the loss reaches and enters the cloud—the yellow area—it signifies that meaningful training is taking place, with the network weights being

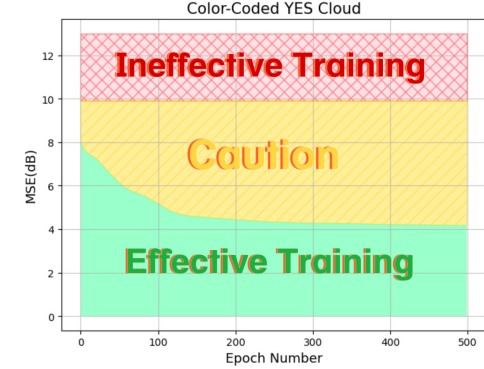

Figure 2: YES training cloud system for quality monitoring: A training loss that remains above the YES training cloud (**red** area) indicates ineffective training. When the loss penetrates the cloud (**yellow** area), it suggests that meaningful training has occurred or is in progress—network weights have been significantly influenced by the data. However, caution is advised, as the training is certainly not optimal. Dropping below the cloud (**green** area) signals effective training in in progress and suggests potential for optimality. It may also indicate diminishing returns in the training process, where further gains could be incremental.

substantially influenced by the data. However, caution is still advised at this stage, as the training has not yet achieved optimality. A loss that descends below the cloud into the green area denotes effective training, suggesting a potential for achieving optimal performance.

## 5 NUMERICAL EXAMPLES AND DISCUSSION

In this section, we numerically assess the effectiveness of our proposed YES bounds in evaluating the performance of the training process for both synthetic and real-world data recovery tasks. For the synthetic data, we generated datasets based on two models: phase retrieval and one-dimensional signal denoising. For the real data example, we applied our bounds to real-world image recovery from noisy quadratic measurements, which is provided in Appendix E.

To generate the dataset for the synthetic data recovery examples, we employ the following models and configurations:

- *Phase Retrieval:* The data is generated by the following model:

$$\mathbf{b}_i = |\mathbf{A}\mathbf{x}_i|, \quad i \in [d], \tag{17}$$

where $\mathbf{A} \in \mathbb{R}^{20 \times 20}$ is a Gaussian sensing matrix with entries drawn from $\mathcal{N}(0, 1/20)$, the signal $\mathbf{x} \in \mathbb{R}^{20}$ is generated as $\mathcal{N}(0, 1)$, and $d = 1000$ samples are generated.

- *One-Dimensional Signal Denoising:* The dataset is constructed using the model:

$$\mathbf{b}_i = \mathbf{x}_i + \mathbf{n}_i, \quad i \in [d], \tag{18}$$

where the signal $\mathbf{x} \in \mathbb{R}^{20}$ is drawn from $\mathcal{N}(0, 1)$, and the noise $\mathbf{n} \in \mathbb{R}^{20}$ follows $\mathcal{N}(0, 0.2)$. For this model, we generate 50 fixed signals, each perturbed by 20 noise vectors sampled from $\mathcal{N}(0, 0.2)$. This process is repeated to generate a dataset of $d = 1000$ samples.

We train five-layer fully connected networks to approximate models under different parameter configurations, utilizing the ADAM algorithm for optimization. In all experiments, the learning rate is initialized at $\eta = \eta_0$ and reduced by a decay factor of $0.9$ every 50 epochs. It is important to note that the bias term is excluded in the phase retrieval model, whereas it is included in the one-dimensional signal denoising task. We give the following criterion for stopping the training: the rate of change in network weights is sufficiently low.

In Fig. 3, we present color-coded clouds illustrating the training process and corresponding bounds for the phase retrieval model under various conditions and parameters. Figs. 3(a)-(c) show the

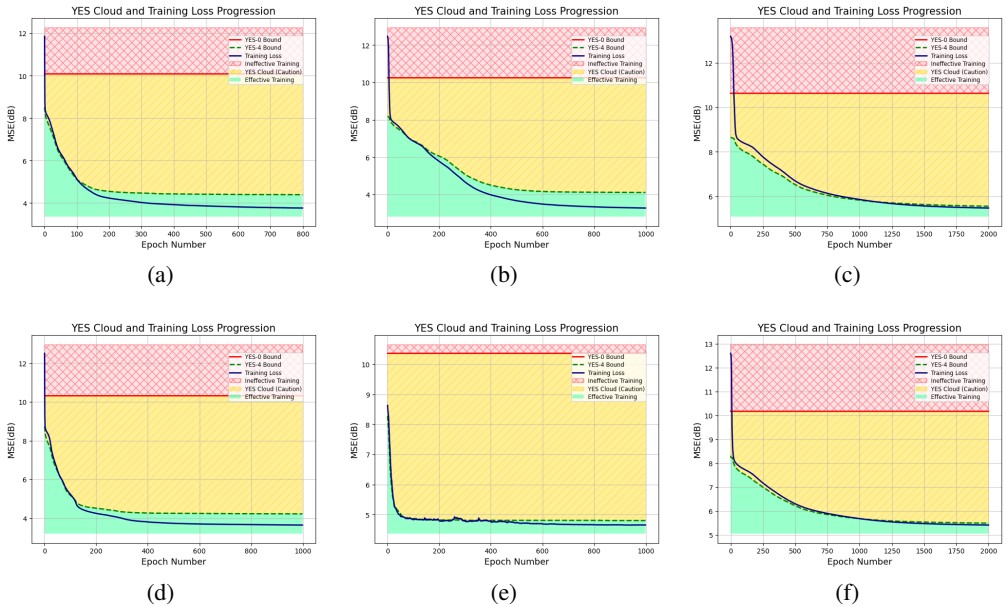

Figure 3: YES training clouds for the phase retrieval model. The clouds are shown for a fully connected network with 5 layers, each corresponding to different training parameter settings. Figs (a)-(c) illustrate the performance of the YES bounds for different batch sizes: 20, 100, and 500, respectively, with a learning rate of $1e - 3$. Figs. (d)-(f) compare the YES bounds to the training process with different learning rates: $1e - 3$, $1e - 2$, and $1e - 4$. As seen in Figs. (b) and (c), increasing the batch size slows the convergence rate, with the training loss entering the green region after more than 100 epochs. Interestingly, when adjusting the learning rate to $1e - 2$ and $1e - 4$, as shown in Figs. (e) and (f), the training struggles to reach the green region, suggesting that a learning rate of $1e - 3$ is the proper parameter for this task. This observation is further supported by comparing the loss functions across Figs. (d)-(f). Another notable observation in Fig. (f) is that both the YES bound and the training loss converge relatively closely until the final convergence, indicating that the training solution behaves similarly to a linear projection.

clouds for different batch sizes, while Figs. 3(d)-(f) illustrate the clouds for varying learning rates. Specifically, Fig. 3(a) corresponds to a batch size of 20, Fig. 3(b) to a batch size of 100, and Fig. 3(c) to a batch size of 500, all with a fixed learning rate of $1e - 3$. Similarly, Fig. 3(d) uses a learning rate of $1e - 3$, Fig. 3(e) uses $1e - 2$, and Fig. 3(f) uses $1e - 4$, all with a fixed batch size of 20.

As seen in Figs. 3(a)-(b), training enters the green region after approximately 100 epochs, signaling effective training is in progress. Notably, the convergence rate for the batch size of 20 is slightly faster than that of 100, as indicated by the earlier entry into the green region. In Fig. 3(c), the model struggles to reach the green region, only doing so after 1250 epochs, suggesting that batch sizes of 20 or 100 are more suitable for this task. Notably, by our real-time bounds, one can notice the ineffectiveness of the chosen parameters in a real-time manner instead of running the model for various parameter settings. In terms of learning rate, Fig. 3(d) shows that a rate of $1e - 3$ leads to effective training, with the loss entering the green region after 100 epochs. In Fig. 3(e), increasing the learning rate to $1e-2$ does not alter the entry point into the green region, but the training loss plateaus closer to the YES bound, implying that the solution aligns with a linear projection—potentially suboptimal in this context. This pattern is consistent in Fig. 3(f) with a learning rate of $1e-4$, where the model enters the green region after 1250 epochs and similarly plateaus near the YES bound. Overall, these observations suggest that $1e - 3$ is a better learning rate for this task compared to $1e - 2$ and $1e - 4$.

In Fig. 4, we present color-coded clouds illustrating the training process and corresponding bounds for the signal denoising model under various conditions and parameters, following a similar approach as in the phase retrieval case. Figs. 4(a)-(c) highlight the effects of varying batch sizes, while Figs. 4(d)-(f) show the impact of different learning rates. Specifically, Fig. 4(a) corresponds to a

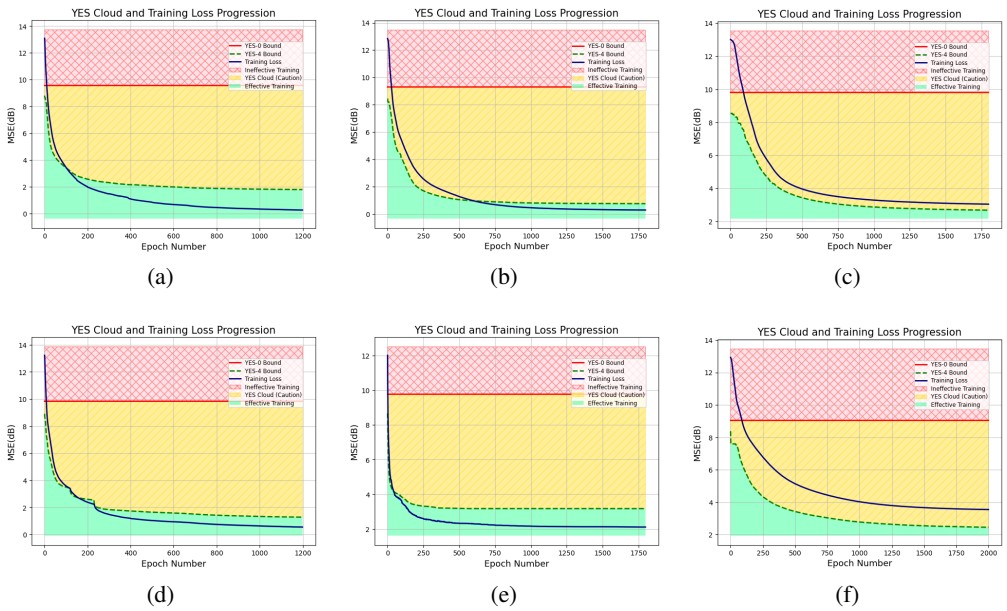

Figure 4: YES training clouds are utilized for the signal denoising task, displayed for a fully connected network comprising five layers, each representing a unique training parameter configuration. Figs. (a)-(c) demonstrate the YES bounds' performance across different batch sizes: 20, 100, and 500 with a learning rate of $1e-3$. Lastly, Figs. (d)-(f) compare the YES bounds against the training process employing varying learning rates: $1e-3$, $5e-3$, and $1e-4$. As shown in Fig. (b), training with a batch size of 100 significantly delays reaching the green region, indicating that convergence is faster with a batch size of 20 than 100. As shown in Fig. (c), the training plateaus in the yellow region, indicating that the solution obtained by the network is far from the optimal. In Fig. (e), increasing the learning rate to $1e-2$ accelerates convergence compared to $1e-3$, while in Fig. (f), the training loss plateaus in the yellow region for the learning rate of $1e-4$.

batch size of 20, Fig. 4(b) to a batch size of 100, and Fig. 4(c) to a batch size of 500, all with a fixed learning rate of $1e-3$. Figs. 4(d)-(f) reflect learning rates of $1e-3$, $5e-3$, and $1e-4$, respectively, using a fixed batch size of 20.

Comparing Fig. 4(a) and Fig. 4(b), we see that the training loss with a batch size of 20 converges faster than with a batch size of 100, as evidenced by the earlier entry into the green region. Although the results of both batch sizes ultimately reach similar error levels, the YES bound for the batch size of 100 is closer to the training loss, a characteristic of our real-time bounds, which are constructed from specific intermediate instances along the optimization path. In Fig. 4(c), the training loss plateaus in the yellow region, indicating that the training is influenced by the data but remains suboptimal, failing to enter the green region.

Turning to the impact of learning rates, Fig. 4(d) shows that increasing the rate to $1e-2$ accelerates convergence compared to $1e-3$. However, in Fig. 4(f), with a learning rate of $1e-4$, the training loss again plateaus in the yellow region, failing to reach the green region and signaling suboptimal performance. These insights into training effectiveness are achieved in real time, without requiring multiple parameter comparisons retrospectively, providing an immediate and clear assessment of the training process. It is important to observe that the bound is more capable than just a local optimality check. For example, you can generate a number of random points around the current weights and plot the minimum loss curve associated with them, noting that for local optimality we should be on that curve. However, one can easily see that the YES bound goes further, as observed in Figs. 4(c),(f), the training losses converge in the yellow.

To check the performance of the YES bounds across different layers, we train fully connected networks for the phase retrieval model, with results shown in Fig. 5. Specifically, Fig. 5(a) illustrates the cloud for a fully connected network with 5 layers, Fig. 5(b) for 6 layers, and Fig. 5(c) for 7 layers. As observed, the convergence rate of 7-layer network appears to slow down under the same

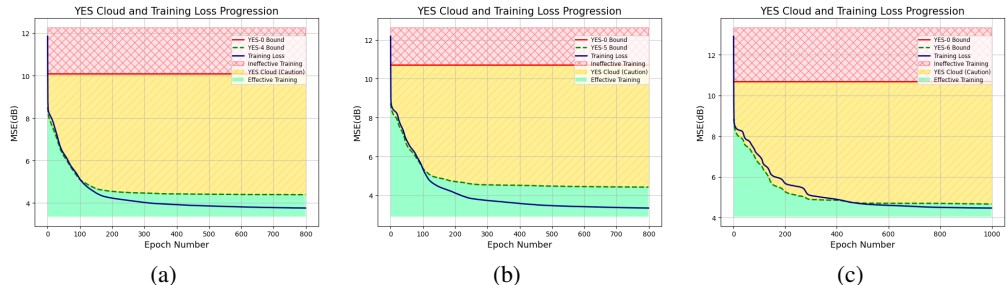

(a)                                (b)                                (c)

Figure 5: YES training clouds for networks with different numbers of layers are shown as follows: (a) 5 layers, (b) 6 layers, and (c) 7 layers. As expected, increasing the number of layers makes the optimization task in the training process more complex. Interestingly, in the case of the 7-layer network, the training loss struggles to reach the green region and stays closely aligned with the result of the YES bound.

parameter settings. For example, in Fig. 5(b), the training takes nearly 100 epochs to reach the green region, whereas in Fig. 5(c), the loss enters the green region after more than 400 epochs. This suggests that real-time monitoring using YES bounds can quickly indicate when the chosen parameters may be suboptimal for the given model architecture.

It is often a challenge for both users and AI professionals to determine the optimal point to stop training (cost vs performance trade-off). A common approach is to monitor the rate of change in the loss function, waiting for the loss to plateau as a sign of potential convergence. However, as shown in Fig. 4 of the Appendix, the training objective of neural networks (and our training procedures) can result in the loss appearing to plateau multiple times before quality training is achieved. This is where the YES clouds come to the rescue: *To know when the smaller rate of change in the loss or the weights of the neural network does not equate optimality, and when to act on it—i.e., when we are in the green.* Additionally, it is insightful to investigate how test results behave as the training progresses through different regions of the color-coded clouds, which is investigated in Appendix D.

## 6 CONCLUSION

One may wonder why the YES bounds work as they do. The reason is simple: heuristics outperform random, and optimal beats heuristic. In mathematics, many effective bounds emerge from insightful heuristics. The YES training bounds are similarly grounded in solid mathematical principles—specifically, that neural networks, with their non-linearities, should outperform linear projections. These bounds provide a useful heuristic that separates the wheat from the chaff, the random from the meaningful. This is particularly evident with the YES-0 bound. But even with the more sophisticated YES bounds and the associated training cloud, the principle holds: they expose the randomness (in the sense of not sufficiently taking the shape of the optimal) in the solution by offering a better real-time heuristic. Ultimately, as the process nears optimality, we witness the final convergence—optimal surpasses heuristic. This transition is marked by entering the green zone beneath the training cloud.

It is worthwhile to note that the YES bounds and their certification should be regarded as a necessary condition for optimality, not a sufficient one. They provide a certificate of potential optimality—a certificate one must certainly have to assert quality in training. They are, however, a concrete step toward bridge the gap between the opaque nature of neural networks and the pressing need for transparency and reliability in high-stakes applications.

Future work may explore the extension of the YES bounds to various network architectures and training paradigms, as well as their integration into automated training systems. Developing more sophisticated bounds and understanding their impact on the AI training ecosystem are promising avenues for research.

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

# DATA-AWARE TRAINING QUALITY MONITORING AND CERTIFICATION FOR RELIABLE DEEP LEARNING

# [SUPPLEMENTARY MATERIAL]

**Anonymous authors**

## A  THE DECREASING BEHAVIOUR OF YES-0 BOUND

**Theorem 1.** *Let $\Omega$ be an activation function in a deep neural network. If $\Omega$ is applied in an element-wise manner and satisfies the following conditions:*

- 1-Lipschitz Condition*:*

$$\|\Omega(\mathbf{x}_1) - \Omega(\mathbf{x}_2)\| \leq \|\mathbf{x}_1 - \mathbf{x}_2\|, \quad \forall \mathbf{x}_1, \mathbf{x}_2 \in \mathbb{R}^n, \tag{1}$$

- Projection Property*:*

$$\Omega(\mathbf{Y}) = \mathbf{Y}, \quad \text{if} \quad \mathbf{Y} \in H_\Omega, \tag{2}$$

*then the YES-0 bound is monotonically decreasing with respect to the depth of the network. That is, for each layer $k$:*

$$\|\mathbf{Y} - \mathbf{Y}_{k+1}\|_{\mathrm{F}}^2 \leq \|\mathbf{Y} - \mathbf{Y}_k\|_{\mathrm{F}}^2. \tag{3}$$

*Proof.* Following our formulations, the error at layer $(k-1)$ is

$$\mathbf{E}_{k-1} = \mathbf{Y} - \mathbf{Y}_k, \tag{4}$$

where $\mathbf{Y}$ is the target output, and $\mathbf{Y}_k$ is the network output after $(k-1)$ layers. At each layer, the network updates its output via:

$$\mathbf{Y}_{k+1} = \Omega(\mathbf{A}_k \mathbf{Y}_k), \tag{5}$$

with $\mathbf{A}_k$ representing the weight matrix associated with the YES-0 bound at layer $k$. The error at layer $k$ is thus:

$$\mathbf{E}_k = \mathbf{Y} - \Omega(\mathbf{A}_k \mathbf{Y}_k). \tag{6}$$

By considering the 1-Lipschitz and projection properties of the activation function $\Omega$, we have:

$$\begin{aligned}
\|\mathbf{E}_k\|_{\mathrm{F}}^2 &= \|\mathbf{Y} - \Omega(\mathbf{A}_k \mathbf{Y}_k)\|_{\mathrm{F}}^2 \\
&= \|\Omega(\mathbf{Y}) - \Omega(\mathbf{A}_k \mathbf{Y}_k)\|_{\mathrm{F}}^2 \\
&\leq \|\mathbf{Y} - \mathbf{A}_k \mathbf{Y}_k\|_{\mathrm{F}}^2.
\end{aligned} \tag{7}$$

Since $\mathbf{A}_k$ is the minimizer of the quadratic criterion $\|\mathbf{Y} - \mathbf{A}_k \mathbf{Y}_k\|_{\mathrm{F}}^2$, we have:

$$\|\mathbf{Y} - \mathbf{A}_k \mathbf{Y}_k\|_{\mathrm{F}}^2 \leq \|\mathbf{Y} - \mathbf{Y}_k\|_{\mathrm{F}}^2 = \|\mathbf{E}_{k-1}\|_{\mathrm{F}}^2. \tag{8}$$

Combining (8) with (7) completes the proof. □

In Fig. 1, we validate Theorem 1 by demonstrating the decay of the YES-0 bound across layers. This result is based on the phase retrieval model.

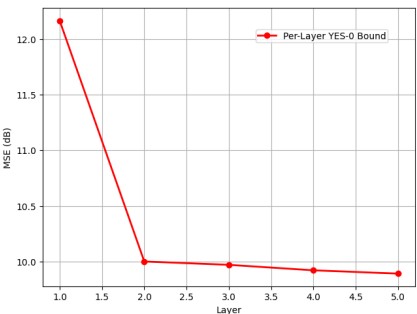

Figure 1: YES-0 decay with respect to the number of layers.

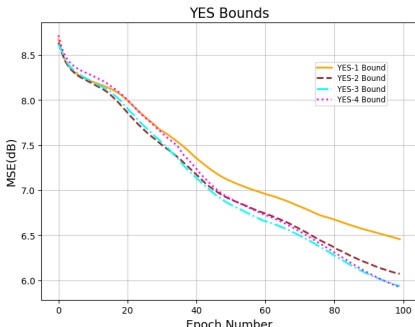

Figure 2: YES training bounds with varying degrees, without imposing monotonicity, are presented. As can be observed, increasing the degree of the bound does not necessarily improve it, as the bounds remain closely aligned with each other.

## B  NON-DECREASING BEHAVIOUR OF YES-$k$ BOUNDS WITHOUT MONOTONICITY

YES training bounds with different degrees, without imposing monotonicity, are shown in Fig. 2 for the phase retrieval model. An interesting observation from this figure is that increasing the degree does not necessarily improve the YES bounds. In fact, all the bounds remain relatively close to each other.

---

**Algorithm 1** Generation of the YES Training Bounds ($k \geq 1$).

---

**Input:** $(\mathbf{X} \in \mathbb{R}^{n \times d}, \mathbf{Y} \in \mathbb{R}^{m \times d})$ are training data matrices with $d$ denoting the number of training samples.
**Output:** YES training bounds.
1: $\mathcal{I} \leftarrow \{2, \cdots, K\}$.
2: $\mathbf{e} \leftarrow \mathbf{0}_{K-1}$.
3: **for** $k = 1 : (K - 1)$ **do**
4:     $\mathcal{H} \leftarrow \text{combination}(\mathcal{I}, k) \triangleright \text{combination}(\mathcal{I}, k)$ is the combination operator that selects $k$ items from the set $\mathcal{I}$.
5:     $\mathbf{u} \leftarrow \mathbf{0}_{|\mathcal{H}|} \triangleright \mathbf{0}_{|\mathcal{H}|}$ denotes a zero vector with the length $|\mathcal{H}|$.
6:     **for** $i = 0 : |\mathcal{H}| - 1$ **do**
7:         $\mathcal{H}_i \leftarrow \mathcal{H}[i] \triangleright \mathcal{H}[i]$ denotes the $i$-th combination item of $\mathcal{H}$.
8:         $l \leftarrow 0$.
9:         $\mathbf{Y}^{\star} \leftarrow [\,] \triangleright [\,]$ denotes an empty tensor.
10:         **for** $j = 0 : (k - 1)$ **do**
11:             $\mathbf{Y}^{\star}.\text{append}(\text{model}_{\mathcal{H}_i}(\mathbf{X})) \triangleright \mathbf{Y}^{\star}.\text{append}(\mathbf{T})$ denotes appending the matrix $\mathbf{T}$ in an empty tensor $\mathbf{Y}^{\star}$, $\text{model}_{\mathcal{H}_i}(\mathbf{X})$ denotes the output of the training model at specific layers specified by the elements in $\mathcal{H}_i$.
12:         **end for**
13:         $\mathbf{Y}_t \leftarrow \mathbf{X}$.
14:         **for** $j = 0 : (k - 1)$ **do**
15:             **while** $l \leq \mathcal{H}_i[j]$ **do**
16:                 $\mathbf{A}_t \leftarrow \mathbf{Y}^{\star}[j]\mathbf{Y}_t^{\dagger}$.
17:                 $\mathbf{Y}_t \leftarrow \Omega(\mathbf{A}_t \mathbf{Y}_t)$.
18:                 $l \leftarrow l + 1$.
19:             **end while**
20:         **end for**
21:         **for** $z = 1 : K - l - 1$ **do**
22:             $\mathbf{A}_t \leftarrow \mathbf{Y}\mathbf{Y}_t^{\dagger}$.
23:             $\mathbf{Y}_t \leftarrow \Omega(\mathbf{A}_t \mathbf{Y}_t)$.
24:         **end for**
25:         $\mathbf{u}[i] \leftarrow \|\mathbf{Y} - \mathbf{Y}_t\|_{\mathrm{F}}^2 / d$.
26:     **end for**
27:     $\mathbf{e}[k - 1] \leftarrow \min \mathbf{u}$.
28: **end for**
29: YES bound $\leftarrow \min \mathbf{e}$.
30: **return** YES bound

---

## C THE YES-$k$ TRAINING BOUNDS ($k \geq 1$) WITH MONOTONICITY

In Algorithm 1, we reformulate the YES-$k$ bounds for $k \geq 1$, incorporating a monotonic modification through the inclusion of YES-$k$ subsets to ensure the bounds remain monotonic.

For the YES bounds with monotonicity, as illustrated in Fig. 3 with various initializations, it is evident that the YES bounds are closely grouped. We investigated this observation using fully-connected networks with both 5-layer and 7-layer architectures, conducted this experiment 1000 times, and consistently observed similar results. This observation suggests that we may leverage the advantages of higher-degree YES bounds by calculating only the first few YES-$k$ bounds, which could be beneficial from a computational standpoint.

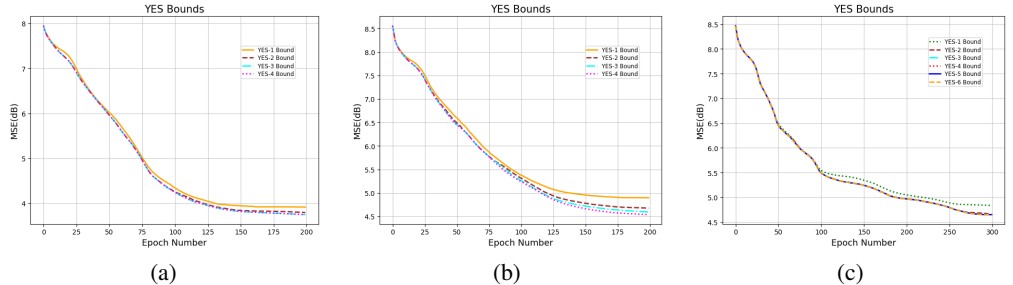

(a)                         (b)                         (c)

Figure 3: YES training bounds with varying degrees, this time incorporating monotonicity across different initializations, are presented. Similar to the non-monotonic case, increasing the degree of the bound does not necessarily enhance it, as the bounds remain close to each other.

## D   TEST RESULTS FOR TRAINING PROCESS MONITORED BY YES CLOUDS

Beyond the training process, it is insightful to investigate how test results behave as the training progresses through different regions of the color-coded clouds. To explore this, we present both training and test outcomes for the phase retrieval model in Fig. 4. As observed, when the training loss decreases in the red region, the test loss similarly declines. When training plateaus in the yellow region, the test loss also plateaus. Interestingly, upon entering the green region, the training loss initially exhibits fluctuations, likely due to the learning rate, before plateauing—a pattern mirrored in the test loss. However, after approximately 2000 epochs, the test loss increases.

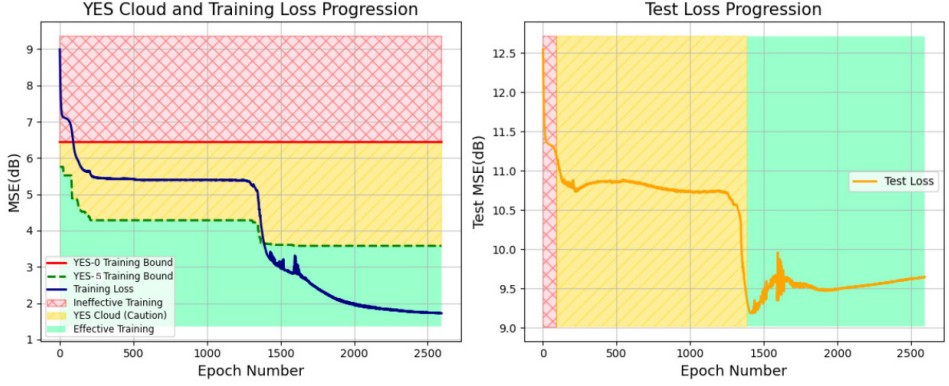

Figure 4: The YES bounds cloud for the training process is presented alongside the test stage for the same training results monitored by the YES training bounds.

# E   IMAGE RECOVERY FROM CORRUPTED QUADRATIC MODEL

To elucidate the practical significance of the YES bounds and their associated cloud system, we further examine an image recovery task for an image degradation process characterized by the model:

$$\mathbf{b}_i = |\mathbf{A}\mathbf{x}_i|^2 + \mathbf{n}_i, \quad i \in [d], \tag{9}$$

where each $\mathbf{x}_i$ represents a distinct patch of the original image undergoing recovery. This model presents a complex degradation process that involves three primary challenges:

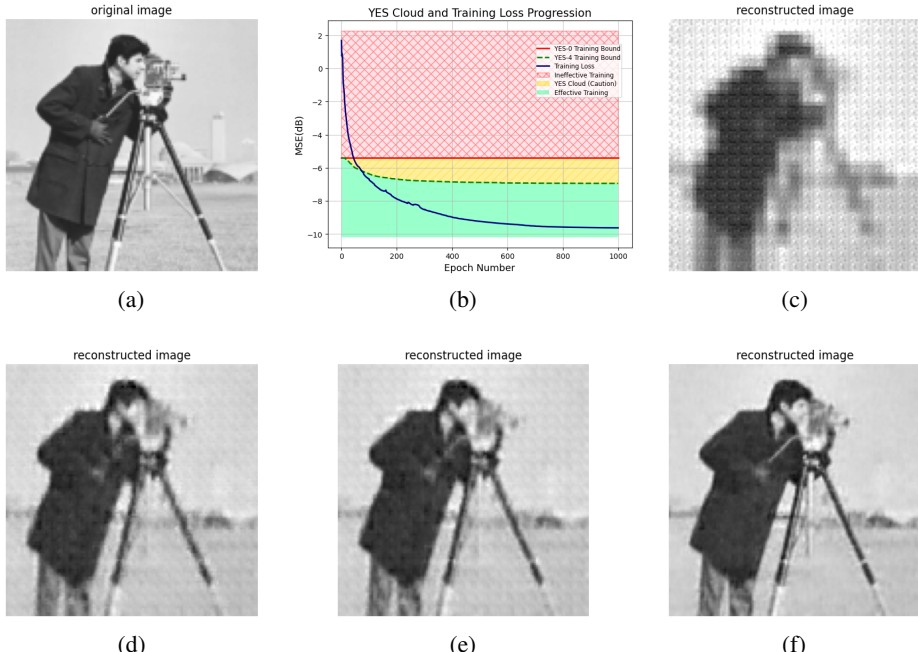

(a)        (b)        (c)

(d)        (e)        (f)

Figure 5: Monitoring the training process for the 5-layer fully-connected network used to reconstruct the cameraman image, as shown in (a), from the corrupted phase retrieval model, with the YES bounds cloud illustrated in (b). The quality of the reconstructed image is presented at different stages of training: (c) at the initial training loss, (d) as the training loss enters the cautionary yellow region, (e) when it reaches the green region, and (f) when the training loss converges to the final solution in the green region. As observed from the reconstructed images, the training performance improves progressively as the loss moves from the yellow region to the green region, achieving the best performance upon convergence. This demonstrates the effectiveness of the YES bounds cloud in monitoring the training process, even for tasks like image denoising.

*1. Blurring Operation ($\mathbf{A}\mathbf{x}_i$):* The matrix $\mathbf{A}$ applies a blurring operator to the image patch $\mathbf{x}_i$, necessitating deblurring techniques to counteract the smoothing effects.

*2. Phase Loss ($|\cdot|^2$):* The absolute value squared operation results in phase loss, requiring phase retrieval methods to restore essential phase information for accurate reconstruction.

*3. Additive Noise ($\mathbf{n}_i$):* The term $\mathbf{n}_i$ introduces additive noise, demanding denoising strategies to mitigate its adverse effects on image quality.

The performance of the proposed YES bounds and cloud system is illustrated in Fig. 5 and Fig. 6, which present the model's recovery performance under four distinct conditions:

- *Training Loss at Initial Value:* At the outset, the training loss is significantly higher than the YES bounds, indicating suboptimal performance. The recovery results at this stage exhibit pronounced blurring, substantial phase distortions, and noticeable noise artifacts, reflecting the model's nascent state.

- *Training Loss at YES-0 (Top of the Cloud):* As training progresses, the loss approaches the YES-0 bound—the top of the cloud. At this juncture, the model achieves a moderate level of recovery, with reduced blurring and phase errors, alongside diminished noise. However, the performance remains below optimal, as indicated by the fact that the training loss has not yet breached the lower bounds of the cloud.

- *Training Loss at YES-$(K-1)$ (Bottom of the Cloud):* Further training brings the loss down to the YES-$(K-1)$ bound—the bottom of the cloud. The recovery results at this stage demonstrate significant improvements, with minimal blurring, accurate phase reconstruction, and negligible noise. This indicates that the model is nearing the performance limits as prescribed by the YES bounds.

- *Optimized Convergence:* Upon convergence, the training loss reaches its optimal value, falling within the YES bounds. The recovery results are exemplary, showcasing precise deblurring, flawless phase retrieval, and excellent noise suppression. This final stage confirms that the model has achieved a state of optimal performance, as validated by the YES bounds.

To numerically validate the training results monitored by the YES bounds cloud, we apply these bounds to a corrupted phase retrieval model using two different images: the $128 \times 128$ cameraman and the boat, where we consider the patch size of $8 \times 8$ of these images for the model in (9). These images help assess the effectiveness of the YES cloud across various regions and determine whether the training loss entering the green region can indeed lead to practical solutions for real-world tasks, such as image denoising. Figs. 5 and 6 (a) show the original image of the cameraman and the boat, respectively, (b) present the YES cloud used for tracking the training loss with the YES bounds, (c) illustrate the initial results when the training loss is in the red region, (d) show the output as the loss enters the cautionary yellow region, (e) depict the outcomes when the training loss reaches the green region, indicating effective training according to the YES bounds, and finally, (f) shows the point at which the training loss converges. As illustrated in Fig. 5(c) and Fig. 6(c), the quality of the reconstructed images at the initial stage of training is poor, with noticeable noise, blurring, and patch artifacts. However, in Fig. 5(d) and Fig. 6(d), these imperfections are reduced compared to the earlier stage. Interestingly, by the time the training loss enters the green region, as shown in Fig. 5(e) and Fig. 6(e), the background of the images appears much clearer, with a significant reduction in noise and blurring artifacts. Finally, once the training fully converges in the green region, we observe a well-reconstructed input version, with most distortions effectively removed.

Note that while dropping below the YES cloud signals entering into the effective training regime, it is certainly not recommended to stop the training once this occurs. In fact, it would be reasonable to continue the training, e.g, as long as the rate of decrease in the loss is satisfactorily large. However, we show in this example that one may expect satisfactory performance even if they stop the training prematurely in the green. Based on this, we recommend the following criteria for stopping the training: 1) in the green, 2) the rate of change in network weights is sufficiently low.

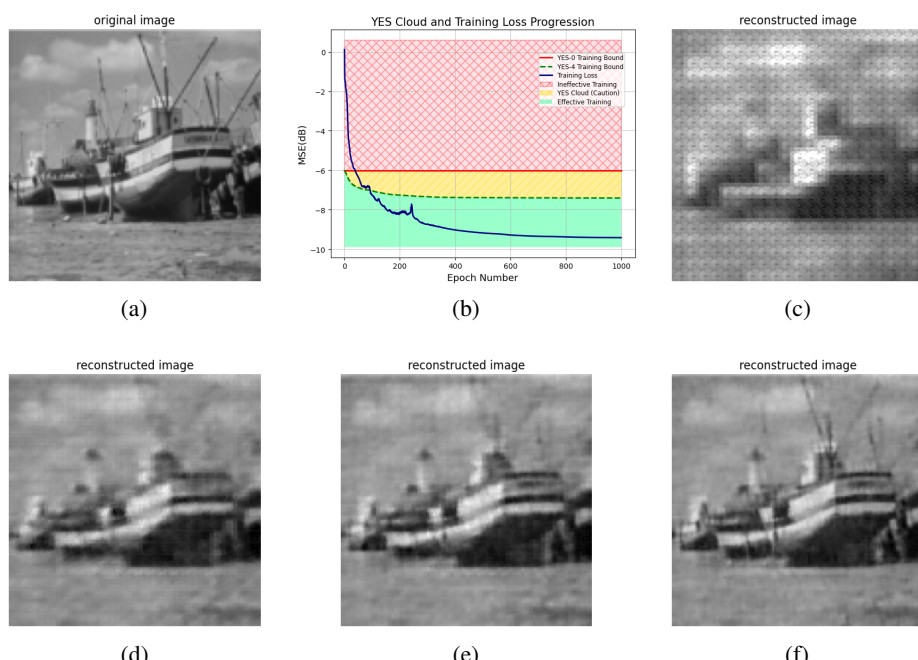

Figure 6: The training process of the 5-layer fully-connected network, which was utilized to reconstruct the boat image from a corrupted phase retrieval model, was closely monitored. The YES bounds cloud, depicted in (b), guided the process. The progression of the image quality through various training stages is showcased: (c) at the initial training loss, (d) as the loss enters the cautionary yellow region, (e) upon reaching the green region, and (f) when the loss stabilizes in the green region, indicating the final solution. The reconstructed images clearly show that as the training loss transitions from yellow to green, the performance steadily improves, culminating in optimal performance at convergence.

## F  MNIST CLASSIFICATION

To further assess the performance of our YES bounds in practical scenarios, we conducted experiments using the MNIST dataset, which was designed for classification tasks. We worked with 5000 training and 5000 test samples. Each image, representing a digit $i \in \{0, \cdots, 9\}$, was encoded by generating a zero matrix with the same dimension as the input image with a single 1 placed at $(i + 1, i + 1)$. A 5-layer fully connected network was trained with SGD, using an initial learning rate $\eta_0$ and a decay factor of $0.7$ every 50 epochs. The classification was performed by minimizing the MSE between model outputs and encoded images, with the success rate determined by accurate classifications over the entire dataset.

As shown in Fig. 7(a), with a learning rate of $1e − 4$, the training loss struggles to move beyond the caution region and remains close to the bottom of the YES clouds. In terms of success rates, Fig. 7(b) displays the training process, while Fig. 7(c) presents the test stage. Although the performance appears satisfactory, the YES cloud suggests that the model's solution is akin to a linear projection, indicating suboptimal training parameters. Adjusting these parameters could lead to improved model performance.

In Fig. 7(d), we apply a learning rate of $5e − 4$ for the solver. In this case, the training loss reaches the green region after approximately 30 epochs. The success rate for the training results, shown in Fig. 7(e), indicates that when the training loss enters the yellow region, the success rate is 85 percent. Once it enters the green region, the success rate increases to 95 percent, and at the convergence point, we achieve 100 percent performance. For the test results depicted in Fig. 7(f), the loss reaches the yellow region at 83 percent, and upon entering the green region, the success rate becomes 92 percent. At the convergence rate, the test results reach 95 percent. As discussed earlier, when the training loss plateaus in the green region, the model's solution can be a strong candidate for optimality.

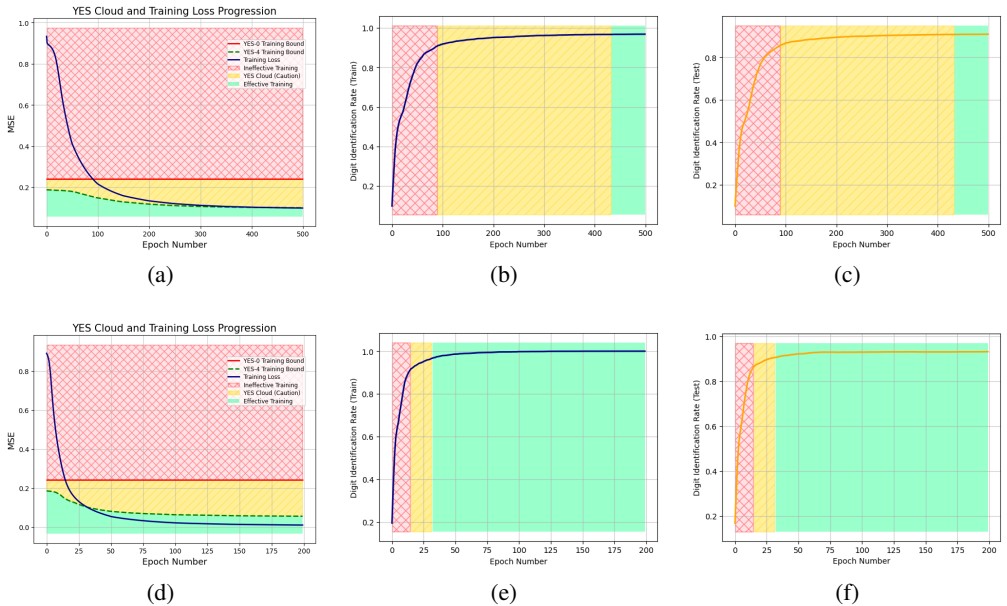

(a)           (b)           (c)

(d)           (e)           (f)

Figure 7: The YES bounds cloud for the training process on the MNIST dataset is presented alongside the success rates for both the training and testing stages. Figs. (a) and (d) show the YES clouds for solvers with different learning rates: (a) $1e-4$ and (b) $5e-4$. Figs. (b) and (c) display the success rates during the training and testing phases, respectively, within the color-coded YES cloud regions. These figures demonstrate how effectively the YES bounds monitor solver performance using a learning rate of $1e-4$. Figs. (e) and (f) illustrate the success rates within the YES cloud regions for the solver using a learning rate of $5e-4$.

Figs. 7(e) and (f) illustrate the model's effectiveness, achieving a $100\%$ success rate in training and $95\%$ in testing. This demonstrates the model's high accuracy and generalization, indicating that it is well-tuned to the task at hand.

## G  BEYOND CERTIFICATION: CAN WE GUIDE THE TRAINING PROCESS?

It is our understanding that the YES bounds and their associated clouds provide not only a mathematical framework for certifying AI training but may also aid practitioners as a method to guide the training process. To harness this dual utility without compromising the integrity of the certification process, it is imperative to maintain a clear information barrier between training and certification. This separation ensures that the training algorithm does not gain access to the YES bound data or the certification network weights. Allowing such access would undermine the certification's purpose by enabling the training process to exploit the bound information, leading to several adverse consequences:

- *Loss of Randomization Benefits:* Randomization, particularly during initialization and throughout training, plays a crucial role in escaping local minima and ensuring robust convergence. If the training process can access YES bound data and network weights, it may inadvertently eliminate these randomization benefits, resulting in deterministic and potentially suboptimal training trajectories.

- *Faulty Optimization Directions:* The training algorithm might adopt step directions that do not align with the true optimization landscape. Since there is no guarantee that the certification network weights resemble the optimum, leveraging these weights could steer the training process in misleading directions, ultimately degrading the quality of the trained model.

- *Obsolescence of the Certification Test:* The primary purpose of the certification test is to provide a reliable bound on the network's training performance. If the training process can consistently operate at or below this bound by utilizing certification data, the test will be rendered ineffective.

To mitigate the risks associated with direct access to certification data, it is essential to devise mechanisms that allow the training process to benefit from the YES bounds without exposing the certification test itself. One effective strategy is to share only the *test results*, such as those visualized through the YES cloud, rather than the underlying certification data or network weights. This approach provides the training algorithm with a lower bound on the distance between the current loss and the optimal loss without revealing any specific information about the certification criteria. Specifically, the distance of the current loss from the bottom of the YES cloud serves as a valuable indicator for adjusting the learning rate:

$$d_k = \max\{\mathcal{L}_k - \mathcal{L}_{\text{YES}}, 0\}, \tag{10}$$

where $\mathcal{L}_k$ is the current loss at epoch $k$, and $\mathcal{L}_{\text{YES}}$ represents the lower bound provided by the YES cloud. This distance $d_k$ may inform the training process on how large of a step size could be chosen to make meaningful progress, ensuring that the learning rate adapts dynamically based on the proximity to the optimal loss. A natural implementation of this guidance mechanism involves defining an adaptive learning rate that incorporates both the traditional vanishing component and an additional term based on the distance $d_k$ to the YES bounds.

