# OpenReview forum: "Data-Aware Training Quality Monitoring and Certification for Reliable Deep Learning"
_ICLR.cc/2025/Conference — ICLR 2025 Conference Withdrawn Submission_

### Official Review · Reviewer_SByb · 2024-10-24

**Soundness:** 2
**Presentation:** 2
**Contribution:** 2
**Rating:** 3
**Confidence:** 3

**Summary:**

The paper explores the challenge of monitoring the quality of training for deep learning models. The paper proposes a method based on upper bounds for the training loss, estimated using linear projections from different layers during training. The authors show that comparing the training loss with the estimated bounds can provide real-time insights into the quality of the training procedure, facilitating the intervention in case of ineffective or sub-optimal training.

**Strengths:**

The idea of using linear projections as a weak model to compute upper bounds on the training loss is interesting and original.

The method is well-described and easy to understand. Mathematical proofs are easy to follow.

Getting insights on the training process beyond the local optimization perspective might be a promising and relevant direction for the future of ML optimization.

**Weaknesses:**

* The paper only analyzes small networks with FeedForward architecture and only Linear layers. It is not clear how the procedure could be applied to practical architectures such as ResNet or Transformers. Additionally, there are no concrete details about the architectures used. I assume all hidden layers have the same dimensionality as the input and output, but this should be clearly stated. I suggest the authors extend their experiments to more complex architectures and to include a detailed description of the architectures used (e.g. a table with the number of layers, number of hidden neurons per layer, activation functions).
* The tasks used for training the networks are not clearly explained. It is not clear what is the input and output of the models, what is the loss function and whether the trained models can achieve any significant generalization power. In particular, for the 1D Signal Denoising task, it is unclear how the random signal drawn from $N(0,1)$ could be recovered after applying noise drawn from $N(0,0.2)$. I would like to see more detailed explanations regarding the tasks and the training setup.
* The practical image recovery experiment in the Appendix does not seem to have real practical applications. My understanding is that the model is specifically trained to recover one image after it has been corrupted, which would require full access to the uncorrupted image during training. If this is correct, then this example has no more practical applicability than the synthetic data tasks. It is also unclear how the network is constructed and what are its inputs and outputs. I suggest adding some more detailed explanations about the training and the model architecture.
* The paper presents no analysis of the running time of the proposed method. It is unclear how the method will impact the training time of the model under real-time conditions. It would be interesting to see a comparison of training times with and without the YES bounds computation and an analysis of how the computational overhead scales with model size (I would expect the costs of estimating higher order YES bounds to significantly increase for very deep models).
* The utility of the proposed method concerning train-test generalization is only presented in the Appendix. I consider that this experiment should be presented in the main body of the paper, as ultimately the purpose of training models is generalization to unseen test data. However, it is unclear how well the model is able to generalize in this case: Figure 4 clearly shows a correlation between the evolution of the training and testing losses, but the minimum value of the test loss is similar to the starting value of the training loss, hinting towards very poor generalization capabilities. This is very likely caused by the low correlation between train and test data, so repeating the experiment on real-world data might show more favorable results. Additionally, this experiment does not present any training details (learning rate, batch size, train/test split ratio etc.).

**Minor points**
* Algorithm 1 should have the model as input along with the training data.

**Formatting**
* The Appendix should be part of the main pdf, not as a separate document
* References to Sections, Figures, Tables, Equations and Citations should have hyper-ref links for better navigability.
* Multiple citations are not well integrated into the text, for example: Line 031 should read “... transformations (LeCun et al.2015, Goodfellow et al. 2016).” which can be achieved by using the \citep{} command. Line 041 should read “Oymak & Soltanolkotabi (2019) theoretically demonstrate …” which can be achieved by using the \citet{} command.
* Line 050 has an unmatched parenthesis after YES.
* Figures 3,4 should have a clear ordering for the learning rates (instead of 1e-3, 1e-2, 1e-4). Subfigure labels for Figures 3,4,5 would also facilitate understanding.

**Questions:**

1. Please address my comments in the Weaknesses section.
2. The experiments show that sometimes the training gets stuck in loss plateaus while in the yellow region. Is there any action that one might take to avoid or overcome this obstacle, other than waiting for the loss to drop?
3. It is not clear how Deep Unfolding Networks are relevant in the context of this paper, as they are only briefly mentioned in section 4.2.1, without sufficient explanation, and then never mentioned again. Can you give a more detailed explanation?

---

> ### Author Response · Authors · 2024-11-23
>
> **W1**: In this work, we introduced data-driven bounds for fully connected networks, representing the first study in this area. Future research could extend our approach to various network architectures, such as CNNs, by applying these bounds to their fully connected layers or adapting them to architectures used in diverse applications, including transformers, as you suggested. Additionally, we tested our proposed bounds on the MNIST dataset and will include these results in the revised manuscript. Based on your feedback, we will also add more comparisons involving different numbers of layers. As you correctly pointed out, this paper assumes that the input and output dimensions are the same. We will emphasize this more clearly in the revised manuscript.
>
> **W2**: To address the reviewer's concern, in phase retrieval model, $\mathbf{X}_i$ and $\mathbf{b}_i$ denote the input and output data, respectively. In the signal denoising model, $\mathbf{b}_i$'s denote the input and $\mathbf{X}_i$'s are the output. The loss function that we considered in all examples was MSE. These examples are synthetic and we just presented them to clearly explain our method and the applicability of the associated YES clouds. However, for the generalization power, we applied our bounds to the MNIST dataset, which we will add in the revised manuscript, and as can be seen, our bound can successfully monitor the performance of digit identification.
>
> **W3**: We totally agree with the reviewer on this since we only considered one image in the training process to show the monitoring power of our bounds in the training regime. However, as we discussed already, we refer the reviewer to see our results on MNIST classification, which is a very common dataset in ML literature.
>
> **W4**: Thank you for your valuable comment. We will include a table comparing the CPU performance for the models we examined in the paper, both with and without the YES bounds.
>
> **W5**: To address your comment, please see the MNIST classification result in the revised manuscript.
>
> **Q2**: Thank you for highlighting this. Please note that our study focuses on monitoring the training process during epochs. While optimization adjustments are beyond the scope of this paper, exploring how YES bounds could be used to modify the optimization process would be an exciting direction for future research. However, in Section~F of the supplemental material, we provided a brief suggestion for this purpose.
>
> **Q3**: We use the Deep Unfolded Network (DUN) as an example to illustrate our point. In a DUN, each iteration of a classical algorithm is unrolled into a network layer, with each layer corresponding to one iteration of the iterative solver. In this context, we expect that increasing the number of iterations will lead to better convergence, meaning that the loss at each layer's output should decrease monotonically, similar to how error decreases with more iterations in traditional algorithms. However, this is not the case with DUNs; they do not exhibit a strictly monotonic behavior.
>
> This suggests that the training process in DUNs might take an indirect path through the optimization landscape to find better solutions, which is similar to the approach we adopted. Additionally, like our model, DUNs assume the input and output dimensions are the same.

---

> > ### Comment · Reviewer_SByb · 2024-11-23
> > **I need to see the revised manuscript**
> >
> > I appreciate the authors' effort in responding to my concerns. However, I cannot change my mind regarding any of my concerns until I have seen the promised new results (e.g. results for training on MNIST or table with performance comparison). Please make use of the functionality of this forum to edit your submission and upload the revised manuscript. I would also want (as Reviewer gBVS also mentioned) to have the appendix included in the main manuscript for easier navigation through the paper.

---

> > > ### Author Response · Authors · 2024-11-25
> > >
> > > Thank you for your thoughtful feedback and for taking the time to reassess our manuscript. We would like to inform you that we have uploaded the revised manuscript. For the reviewers' convenience, we have included the appendix in the main paper. The MNIST classification results can be found in Section F of the appendix.
> > >
> > > Thank you for your consideration, and we welcome any further questions or suggestions you may have.

---

> > > > ### Comment · Reviewer_SByb · 2024-11-27
> > > > **Reply to Authors (1/2)**
> > > >
> > > > I thank the authors for responding to my concerns and for uploading the revised version of their manuscript along with new experimental details. Below I will discuss my previous concerns and the extent to which they were addressed.
> > > >
> > > > **W1**: I appreciate the effort to run additional experiments on a more reality-grounded setting with the MNIST dataset. The details provided in the revised manuscript show that the method can be applied to a more realistic scenario. However, the new experiments bring me some more questions:
> > > >
> > > > **W1.1**: I understand that to suit the experimental setup of the YES bounds, the output of the classification problem has to be transformed into a matrix with the same size as the input, instead of consisting of a 10-dimensional vector. Why was this necessary? From my perspective there should be no problem to have the last layer as a linear projection to a lower dimensional space. If this limitation cannot be easily addressed within the YES bounds framework, it may indicate challenges in adapting the method to broader scenarios.
> > > >
> > > > **W1.2**: While the results obtained with only 5000 samples for train and test are still valuable, I don’t understand why this down-scaling is necessary. MNIST is one of the smallest image datasets, and training a network that gets over 90% test accuracy can be done in less than 1h even without GPU resources. From my perspective this points toward some scalability issues with the method. Could the authors comment on this?
> > > >
> > > > **W1.3**: While the YES bounds can be used to correctly identify a better learning rate, it would be helpful to clarify their distinct advantages compared to simpler heuristic approaches, such as analyzing early learning curves. From the provided plots, it can be easily inferred that 5e-4 is a better value for the learning rate even after 10-25 epochs.
> > > >
> > > > **W1.4**: The YES bounds identify a green region toward the end of training even with a poor choice of learning rate. However, the train/test performance of the network does not seem to be in a good state. One might think that when reaching the green area the training goes well. How should this be interpreted?
> > > >
> > > > **W2.1**: I am even more confused after reading the response. Naming the inputs with X and the outputs with b in one case just to revert them in the other does not seem appropriate. The revised manuscript could benefit from clearer and more consistent notation to reduce potential confusion.
> > > >
> > > > **W2.2**: The authors have also not provided an explanation of how the signal recovery is expected to work in the 1D Signal Denoising task, given the high noise level and the ambiguous (random?) nature of the signal. I don’t understand whether there exists a mathematical function that can map the noisy signal to the original signal in this artificial setup. In the context of digit classification, the main assumption is that there exists a mathematical function composed of matrix multiplications and non-linear activations which can map the image to the corresponding digit. Moreover, there is an assumption that the inputs to the function follow some non-uniform and low-entropy data distribution (because with very high probability sampling an image with random pixels does not yield any recognisable digit). In this case, the notion of denoising (e.g. by diffusion networks) is usually formulated as finding a sample that is close to the noisy sample, while also having high probability in the original data distribution. In the presented artificial experiments, the underlying signal distribution is just a normal distribution, having high entropy and therefore I don’t understand how the signal denoising can be formulated.
> > > >
> > > > **W3**: Addressed by adding the MNIST experiments.
> > > >
> > > > **W4**: The promised table is not present in the revised manuscript. I feel this is even more important when considering the **W1.2** question I raised about the MNIST experiments. I know it might be too late to upload a new revised manuscript, but you can still provide the table in a markdown format in your next response.
> > > >
> > > > **W5**: Partially addressed by the MNIST experiments. I still argue that it would be more relevant to have the practical experiments in the main paper rather than the appendix.

---

> > > > ### Comment · Reviewer_SByb · 2024-11-27
> > > > **Reply to Authors (2/2)**
> > > >
> > > > **Q2**: I still consider that more thorough insights and experiments would be needed to be able to affirm that the YES bounds can actually be useful in adjusting the training process. The discussion in Appendix F (now Appendix G in the revised manuscript) is rather vague and could be strengthened by providing concrete recommendations or examples of how the proposed distance indicator could be effectively used to select hyperparameters in practice. To be convinced by the utility of the YES bounds, I would like to see at least one proposal for a method to choose hyperparameters that works when the trivial methods (such as simply monitoring the performance in th first 10 epochs for different hyperparameter values and greedily choosing the best one) fail to provide the choice that leads to the best final performance.
> > > >
> > > > **Q3**: I feel the analogy with the DUN networks is not very clear. My basic understanding (which might be far from complete) from the short description in the paper that a DUN is conceptually similar to a transformer. They both have a list of blocks which process the information while not changing its dimensionality. In the case of a transformer, the most relevant information can be usually collected either at the first layer (word embeddings) or at the last layer (where full contextual embeddings should have formed). Any layer in between may or may not contain easily processable and therefore useful information. This would be in contrast to the analogy with the training process: during training by gradient descent, the loss is on average decreasing with each iteration, and in theory the network obtained should be better at approximating the function that maps inputs to outputs. If we want a non-monotonic training process we need to consider alternatives to gradient descent, such as simulated annealing.
> > > >
> > > > **Comment**
> > > >
> > > > While the proposed idea is interesting and the additions made in the revision are valuable steps forward, further work is necessary to fully establish the utility and generalizability of the method. Addressing the concerns raised here – such as scalability, clarity in notation, and providing actionable insights for practitioners – would make the paper much stronger and more impactful for the ML community. I encourage the authors to continue refining their approach, as it has the potential to make a meaningful contribution to the field. Considering all of the above, my revised score would be closer to 3.5-4, but the platform restricts me to choose 3.

---

### Official Review · Reviewer_cAaf · 2024-11-02

**Soundness:** 2
**Presentation:** 2
**Contribution:** 2
**Rating:** 5
**Confidence:** 2

**Summary:**

This paper introduces YES training bounds, a framework for real-time, data-aware certification of training, which aims at assessing the quality of model training. Specifically, these bounds evaluate the efficiency of data utilization and optimization dynamics. The depth and non-linear activation functions of models are taken into consideration. Experimental validation on synthetic and real data demonstrates the effectiveness of the bounds in certifying training quality.

**Strengths:**

1. The discussed topic is interesting, which focuses on monitoring the training quality and progress. With the proposed bounds, users could better control the optimization, which could benefit the community.
2. By considering the specific structure and properties of training data, the proposed YES bounds could provide tailored and precise evaluations of training performance.
3. Some experiments on image denoising task demonstrate the effectiveness of the proposed bounds.

**Weaknesses:**

1. No detailed discussion of relevant works, which makes it difficult to situate this paper. Some discussions of relevant optimization works are missing, such as [a, b].
2. The contribution of this paper could be overclaimed. YES bounds are introduced to indicate the training quality. Although the authors claim that the proposed training bounds aim at improving the reliability, robustness, and safety of models, it is difficult to see how the YES bound can be utilized for such a purpose.
3. This paper discusses the scenario of non-direct paths in Section 4.2.1, however, it is difficult to see how the enhanced bounds that involve intermediate points can tackle this issue.
4. Another concern lies in the evaluation, it is unclear whether the proposed training bounds can be generalized to different CV and NLP applications, such as image generation via diffusion, VQA via LLaVa, etc.
5. It is also unclear how the proposed training bounds can motivate new research works or provide insights into this field.

[a]. Generalization Bounds for Stochastic Gradient Descent via Localized ε-Covers. NeurIPS 2022.

[b]. Closing the convergence gap of SGD without replacement. ICML 2020.

**Questions:**

1. Please discuss the relevant work in the field of optimization and clarify the novelty of this paper.
2. Please clarify the contribution and explain the application of YES bounds in the field of model robustness and reliability.
3. Please explain why the enhanced bounds in Section 4.2.2 can tackle the issues discussed in Section 4.2.1.
4. Please provide more evaluation on different CV/NLP tasks to highlight the generalization of the proposed bounds.

---

> ### Author Response · Authors · 2024-11-23
>
> **W1**:  Thank you for highlighting this. We will cite these works and emphasize their contributions in the introduction of our revised manuscript. However, please note that our study focuses on monitoring the training process during epochs. In Section F of the supplemental material, we provided a brief suggestion for optimization adjustments, but further investigation is beyond the scope of this paper and would be an exciting direction for future research.
>
> **W2**: We appreciate the reviewer’s insightful comments. We would like to highlight that model safety and robustness have multiple key elements, with training quality being a cornerstone that significantly influences reliability. If the training performance is poor, it is unreasonable to expect meaningful or reliable results from the model. In this paper, our primary objective is to monitor the quality of the training process. Leveraging our proposed mathematically grounded framework, we introduce the YES bounds and their associated cloud system, which serve as a sanity check for the optimizer. While we agree with the reviewer that we did not exhaustively show the impact of our bounds on test results, we have results (Fig 4 of Supplemental material) that show the correlation between training quality and test outcomes. We will further include MNIST test results for both training and test stages in the revised manuscript. We appreciate your comments bringing to our attention that without clarifying how central training quality is in model reliability, the contribution may be understood as over-stated. We hope that our clarifications have been helpful. Thank you!
>
> **W3**: By direct path, we refer to the scenario that we progressively project the input of each layer to the output $\mathbf{Y}$. In our proposed framework, YES-k, we chose a non-direct path approach which selects the intermediate mapping points $\mathbf{Y}_k$ from the training process, itself. Our motivation for this is that it has been known in ML venue that when training minimizes the objective function in Eq.~(1) in the main paper, then we probably achieve meaningful intermediate mapping points that minimizes the objective greatly. Therefore, we anticipated that if we choose these mapping points from the intermediate layers of the training itself, we may hope for a tighter bound compared to YES-0. In the numerical results, we can observe this very phenomenon which indicates that as training improves it is able to provide enhanced intermediate mapping points for our YES training bound machinery.
>
> **W4**: In this work, we introduced data-driven bounds for fully connected networks, marking the first study in this area. Future research could extend our approach to various network architectures, such as CNNs, by applying these bounds to their fully connected layers or extend these bounds to architectures used in various applications, such as NLP and diffusion models, as you mentioned. Please note that we have an evaluation of our proposed bounds on the MNIST dataset, which will be added in the revised manuscript.
>
> **W5**: We have a research roadmap to extend our framework to apply it to CNNs, DUNs (deep unfolding networks), and PINNs.

---

> > ### Comment · Reviewer_cAaf · 2024-11-25
> >
> > I thank the authors for their response. However, most of my concerns have not been well addressed. For example, it is still difficult to see the clear evidence that YES bound can contribute to the reliability, robustness, and safety of models. The results (Fig 4 of Supplemental material) are not convincing enough. The provided evaluation is rather limited, which is inconsistent with the claims made by the authors. Thus, I maintain my score.

---

> > > ### Author Response · Authors · 2024-11-25
> > >
> > > Thank you for your thoughtful feedback and for taking the time to reassess our manuscript. We appreciate your acknowledgment of the interesting aspects of our work and understand that our previous response may not have fully addressed your concerns.
> > >
> > > --- Allow us to clarify: The reliability and robustness of a machine learning model are fundamentally rooted in the quality of its training process. A model trained under suboptimal conditions is more likely to perform unpredictably or fail when exposed to new data. By providing a framework to monitor and certify the training quality in real-time, the YES bounds help ensure that models are being trained effectively, which is a crucial step toward building reliable and robust systems.
> > >
> > > --- We understand your concern regarding the limited scope of our experimental evaluation and the convincing power of Figure 4 in the supplemental material. In response to your feedback, we have conducted additional experiments on the MNIST dataset (as can be seen now in the revised manuscript), a standard benchmark in machine learning. These experiments demonstrate that the YES bounds effectively monitor training quality in a classification task and correlate with both training and test performance.
> > >
> > > --- Our approach is grounded in solid mathematical principles from optimization theory. This theoretical robustness suggests that the YES bounds have general applicability across different models and training scenarios. While extensive empirical evaluations are valuable, we hope you agree that a strong theoretical foundation provides a more enduring contribution to the field.
> > >
> > > --- Our work serves as a pioneering step toward a new line of research focused on training quality, which has often been overshadowed by emphasis on generalization.
> > >
> > > --- We believe that for reasons that include our initial phrasing of the response the impact of the work was underestimated. We hope this detailed explanation addresses your concerns and demonstrates the relevance and potential impact of our work on the reliability, robustness, and safety of machine learning models.
> > >
> > > Thank you for your consideration, and we welcome any further questions or suggestions you may have.
> > >
> > > Sincerely,
> > >
> > > The Authors

---

### Official Review · Reviewer_gBVS · 2024-11-03

**Soundness:** 1
**Presentation:** 2
**Contribution:** 1
**Rating:** 3
**Confidence:** 4

**Summary:**

The paper presents a heuristic for convergence diagnostics in multi-layer perceptrons that is data-aware. The authors propose to use solve a OLS-like linear problem for each layer to determine what a reasonable, but suboptimal weight matrix for each layer is. They then propose a traffic light system that compares training loss to this heuristic.

**Strengths:**

S1: Convergence diagnostics is a worthwhile problem to study. The criticism of local, curvature based convergence diagnostics is justified and hence, providing a empriical, yet strategic "benchmark" framework is an interesting approach.

S2: The proposed method is simple and readily accessible, even to a non-specialist audience, that is likely to benefit most from "training support".

**Weaknesses:**

**Motivation**

W1: I am sceptical of the utility of a "standardization of training practices in deep learning" (L078). The author's more detailed account that "The proposed YES [is] a promising pathway toward establishing a benchmark for the AI industry, regulators, and users alike" (L070) is vague and does not provide concrete details on _how_ this is valuable. For instance, the authors could provide examples or consult recent legislation on this issue.

W2: Prior work is not properly cited. In fact, only a single paper (Oymak et al.) on neural network optimisation / convergence diagnostics is cited.

**Presentation**

W3: The Appendix is missing.

W4: The language and notation for the main results is not always clear. Examples:

- Eq 3 uses $\mathbf{Y}_k$ without ever defining it before. I suggest the authors provide a clear definition first.
- First paragraph in Section 4.1 is not clear (L183 - 189). Same for L245-L253.
- The paper describes a theoretical optimal model, the actual model, and the "bounding" model obtained by setting weights through the pseudoinverse. It is not always clear which of these models weight matrices or activations belong to.
- It is not clear what "intermediate states", "intermediate mappings", "intermediate points" are.

W5: The authors are inconsistent in their claims and tend to overstating their contributions. While the authors state at times that their method is a "sanity check", they later state:

- "[...] can attest that the training is not proper." (L205)
- "The answer (YES or NO) will provide immediate relief as to whether training has been meaningful at all" (L211)
- "The reason is simple: heuristics outperform random, and optimal beats heuristic" (L521) or L534 to L536.
- "These bounds aim to provide a qualified answer to the question as to whether a neural network is being properly trained by the data: YES or NO?" (L179).
- "cloud unequivocally indicate suboptimality" (L085). This is a very strong statement, for which there is no supporting evidence.
A more precise account of the contributions and limitations would be appropriate as well as fewer absolute statements without justification.

W6: The proposed method is a _heuristic_ not a certificate, which typically describes provable statements that can be asserted about a model.

W7: The authors are overstating the impact to the safety of models. This work predominantly cites literature on ML safety, but does not set out a clear path how their work impacts safety, how it can establish trust and how it will help regulators. The statement "This standardization could play a crucial role in fostering trust and accountability within the AI ecosystem." lacks proper justification.

**Method**

W8: Using a linear model as baseline during training as bechnmark is common practice (when appropriate). The main insight seems to be the prediction of the target from each intermediate layer. I disagree with the authors statement: "A sensible but sub-optimal approach" (L185) and do not see sufficient justification for this statement. The authors state: "it has also been observed in various machine learning problems that after extensive training (resembling what we can describe as optimal training), the output of some inner layers become something meaningful to domain experts" (L255-258), for which they do not provide sources.

W9: layerwise OLS solutions are a very basic heuristic that do not mark significant contributions to the ML community. One can trivially, see that this bound becomes vacuous, even for single layer models when replacing ReLU with sigmoid (i.e. regard data-generating model $\sigma(AX+e)$ where $A$ has large values.)

W10: At times, the authors do not provide sufficient proof or citation when making non-trivial statements.

- "Given a judicious selection of ... the latter should provide a tighter error bound compared to the YES-0 bounding approach" (L252). What is a judicious selection? Such claim should be supported by a theorem or a more detailed analysis.
- L255-258 as cited above.

W11: The authors solely focus on the train loss, when in practice the test loss is most relevant in learning problems.

**Experiments**

W12: The paper describes experiments on 2 small toy data. This is not enough to extrapolate to real impact. There are various regression and classification datasets that seem like suitable tests for this method. In particular, Boston / California Housing Prices, SVHN, CIFAR-10(0) or TinyImageNet. Convergence diagnostics will become more relevant the more complicated the loss landscape and the more non-linear the problem gets. At the same time the proposed bound will become more vacuous in these settings. A detailed discussion of this would be of interest. Larger, more complex, and high-dimensional datasets are important to judge the potential impact of this method.

W13: The authors do not compare their method against diagnostics baselines. For instance, fitting a simple one-layer linear model, or discussing other strategies for convergence diagnostics.

W14: Figure 3 stops before models are converged.

**Minor**

MW1: It is community standard to have pdf links between in-text citation and the bibliography. This would be appreciated. Citations should be in round parantheses when not not part of the sentence's grammatical structure, e.g. (Goodfellow et al., 2016).

MW2: Figure 3: should share X and Y axis.

MW3: The lack of algorithm boxes, makes it difficult to follow the exact procedures described.

**Questions:**

Q1: Could the authors please clarify L090-L092: "They do not produce varying certification results across different training realizations, even when initialized identically or following similar optimization paths."

Q2: Why would randomness (L088-L095) be such an issue?

Q3: $\mathbf{Y}_{k}$ is never defined. What does it denote?

Q4: Could the authors clarify L245-L253. How does one obtain the YES-SIGMA bound precisely?

Q6: How will the YES cloud help regulators?

Q7: Is there a formalism to justify why projecting from each layer to $Y$ is reasonable?

---

> ### Author Response · Authors · 2024-11-23
>
> **W1**: We appreciate the reviewer's comment. The focus of this work is on reliable training, which serves as a necessary condition for reliable test performance. To address the reviewer's concern, we will provide the example of California legislation on the importance of safe AI:
> California is considering several legislative measures to regulate artificial intelligence, focusing on advanced AI systems' safety and accountability. One of the most significant is Senate Bill 1047 (SB 1047), which aims to establish safety and compliance standards for developers of powerful AI models, particularly those with the potential to cause significant harm.
>
> Key Provisions of SB 1047:
> 1 - Independent Audits: AI developers must annually engage third-party auditors to evaluate their compliance with safety protocols, including assessing potential risks and reporting any non-compliance to the California Attorney General​
>
> 2- Risk Assessments: Companies are required to perform pre-deployment risk evaluations and submit detailed compliance reports annually, outlining any critical harm their models might cause and the safeguards in place​.
>
> 3-Incident Reporting: Any significant safety incidents must be reported within 72 hours​
>
> 4-Computing Cluster Regulations: Operators of large-scale computing systems must monitor AI training activities, verify customer identities, and retain data to ensure compliance. They are also required to shut down non-compliant AI training activities if necessary​
>
> Legislation like this highlights a growing emphasis on regulatory frameworks for AI safety. Tools such as the YES training cloud could play a pivotal role in such contexts. For instance, regulators might introduce a training quality compliance clause with requirements such as: “AI developers must ensure training quality by demonstrating that the model’s training loss has plateaued within the green region of the YES training cloud. This evidence must be documented and retained for inspection, particularly in applications involving public utility.” This demonstrates the relevance and potential impact of reliable training in meeting emerging regulatory demands and underscores the broader significance of our work.
>
> **W2**: Thank you for this comment. We will add as many as possible relevant references in the revised manuscript.
>
> **W3**: We submitted the Appendix as a separate file and we apologize for any confusion this may have caused.We will integrate it into the main paper.
>
> **W4**:
> - Thank you for bringing this to our attention. We apologize for overlooking the inclusion of the notation's definition prior to Equation~3. $\mathbf{Y}_k$ means the output of $(k-1)$-th layer.
>
> - We appreciate the reviewer’s comment. In this context, by "A sensible but sub-optimal approach," we refer to a heuristic method that one can utilize to optimize the weight matrices. We say it is sensible since the main goal of the network is to project the input to $\mathbf{Y}$ if there is no any side information of the intermediate mapping points. The reason for suboptimality is (i) we progressively project the input data $\mathbf{X}$ into the output data $\mathbf{Y}$ in each layer, and (ii) we (almost) overlook the non-linearity $\Omega$ and obtain the weight matrices following Eq.~(10). Please refer to our response to comment W10, where we provide clarification on the reasoning presented in L245-L253.
>
> - In our paper, the bounding model is used to evaluate the YES bounds. To clarify this point, we will add a further explanation in the revised manuscript.
>
> - To address the concern raised by these words, please note that all these words refer to $\mathbf{Y}_{k}$, the output of $(k-1)$-th layer.
>
> **W5**: The purpose of a sanity check is to assess whether the approach—in this case, the training process—is functioning as expected. We believe all these statements refer to the sanity check using the YES bounds, which are used to monitor the training process's performance and verify its effectiveness. When we refer to the sub-optimality indicated by our color-coded cloud, we mean that if a solution falls within the red or yellow regions, it signifies the existence of better solutions within the optimization landscape, for instance the solutions provided by YES-0 and YES-$k$. This implies that such solutions are not optimal.
>
>
> .

---

> ### Author Response · Authors · 2024-11-23
>
> **W6**: Heuristic methods in the optimization are sensible and sub-optimal approaches, especially when exact analytical or approximate solutions are unavailable or when a baseline comparison is needed. To evaluate whether an iterative algorithm or non-convex optimizer can outperform these heuristics, we compare their performance against what we call "YES bounds"—reference solutions derived from heuristics. If the training process fails to surpass the heuristic benchmark, it indicates that the optimization approach may be insufficient or sub-optimal. Therefore, this comparison helps certify whether the training outcome is effectively optimized or remains suboptimal relative to the provided heuristic solution.
>
> **W7**: We appreciate the reviewer’s insightful comments. We would like to highlight that model safety and robustness have multiple key elements, with training quality being a cornerstone that significantly influences reliability. If the training performance is poor, it is unreasonable to expect meaningful or reliable results from the model. In this paper, our primary objective is to monitor the quality of the training process. Leveraging our proposed mathematically grounded framework, we introduce the YES bounds and their associated cloud system, which serve as a sanity check for the optimizer. While we agree with the reviewer that we did not exhaustively show the impact of our bounds on test results, we have results (Fig 4 of Supplemental material) that show the correlation between training quality and test outcomes. We will further include MNIST test results for both training and test stages in the revised manuscript. We appreciate your comments bringing to our attention that without clarifying how central training quality is in model reliability, the contribution may be understood as over-stated. We hope that our clarifications have been helpful. Thank you!
>
> **W8**:  We appreciate the reviewer's insightful comment. At first we want to point out that our baseline model is not linear. It is true that in order to obtain the weight matrices layer-wise, we overlooked the effect of activation function $\Omega$. However, later the effect of $\Omega$ has been considered in obtaining the intermediate mapping points to respect the architecture of the network as the reviewer can notice in Eqs.~(12) and (14) in the main paper. Regarding the reviewer's comment "A sensible but sub-optimal approach", we would kindly refer the reviewer to our response to the second part of W4.
>
> **W9**: We appreciate the reviewer's comment. We should highlight that layerwise OLS solutions are only one component of our framework. The overall machinery is shown to work in the provided examples. Additionally, we note that this is a pioneering work in the arena of non-randomized training quality certification from an optimization perspective, and in fact, we are currently working on more sophisticated projections to enhance the bounds (balancing certification quality and computational cost of certification).
> In the revised manuscript, We are including our evaluation of the YES trainingbound framework on MNIST dataset which has produced very interesting results. Regarding your comment "one can trivially, see that this bound becomes vacuous, even for single layer models when replacing ReLU with sigmoid (i.e. regard data-generating model $\sigma(\mathbf{A}\mathbf{X}+\mathbf{e})$ where $\mathbf{A}$ has large values.", we should note that for Sigmoid activation function, our YES bounds framework will slightly change. Note that in obtaining our bounds, we utilized the fixed-point property of $\Omega$,
>
> $\|\mathbf{Y}-\Omega(\mathbf{A}\mathbf{X})\|_{\mathrm{F}}^2$
>
> $=\|\Omega(\mathbf{Y})-\Omega(\mathbf{A}\mathbf{X})\|_{\mathrm{F}}^2$
>
> $\leq\|\mathbf{Y}-\mathbf{A}\mathbf{X}\|_{\mathrm{F}}^2$,
>
> where in the last step we have utilized the 1-Lipschitz property of the ReLU. Therefore, by minimizing the upper bound with $\mathbf{A}=\mathbf{Y}\mathbf{X}^{\dagger}$, we can minimize the left-hand side of the equation. To extend this result for the Sigmoid activation function $\sigma$, we know that $\sigma$ is one-to-one map. Therefore, there exists $\mathbf{Y}^{\prime}$ such that $\mathbf{Y}=\sigma(\mathbf{Y}^{\prime})$. As a result, we can write
>
>    $ \|\mathbf{Y}-\sigma(\mathbf{A}\mathbf{X})\|_{\mathrm{F}}^2$
>
> $=\|\sigma(\mathbf{Y}^{\prime})-\sigma(\mathbf{A}\mathbf{X})\|_{\mathrm{F}}^2$
>
> $\leq\|\mathbf{Y}^{\prime}-\mathbf{A}\mathbf{X}\|_{\mathrm{F}}^2,$
>
>  where in the last step we have utilized the 1-Lipschitz property of the Sigmoid. Therefore, for the Sigmoid activation function, by minimizing the upper bound with $\mathbf{A}=\mathbf{Y}^{\prime}\mathbf{X}^{\dagger}$, we can minimize the left-hand side of the equation. Clearly, this modified bounding approach for Sigmoid does not become vacuous for large values of $\mathbf{A}$.

---

> ### Author Response · Authors · 2024-11-23
>
> **W11**: We thank the reviewer for this insightful comment. We completely agree with you on the importance of investigating the performance during the test stage. We would like to highlight that model safety and robustness have multiple key elements, with training quality being a cornerstone that significantly influences reliability. In the race to make AI systems more reliable, generalization has taken center stage. But what about ensuring the reliability of the training process itself? To address this very question, this paper focuses on performance during the training stage. Clearly, good training performance is a necessary condition for achieving good test results. That said, we do have results (Fig 4 of Supplemental material) that show the correlation between training quality and test outcomes. We will further include MNIST test results for both training and test stages in the revised manuscript.
> It would be highly interesting for future work to further link our training monitoring approach with test-stage monitoring schemes. We have currently included test results alongside the training process to illustrate how test performance varies across different cloud regions.
>
> **W12**: Thank you for this insightful comment. We completely agree with you on this point. We applied our YES bounds to the MNIST dataset for a classification task to further evaluate our proposed scheme on real-world and larger datasets. The results is included in the revised manuscript to demonstrate the effectiveness of our method better. In the tasks examined in this paper—MNIST, image denoising, and the provided synthetic data—we observed that our bounds remain relevant and effective for assessing the performance of the optimizer. For tasks requiring a sigmoid activation function, we kindly refer you to our response to W9, where we address your concerns about the validity of our bounds in such scenarios.
>
> **W13**:  Please note that convergence diagnostics consist of two main steps: monitoring the training process and modifying the optimization process if it fails to converge or moves toward suboptimal solutions. In this paper, we focus on monitoring the training process using a color-coded cloud based on the YES bounds. The modification of the training process falls outside the scope of this work at this stage and can be explored in future research.
>
> **Q1,2**: Since we are proposing a bound for training process, the intrinsic definition of a bound implies that it should not vary if the initialization and the optimization path are the same (fixed).
>
> **Q3**: We refer the reviewer to our response in W4.
>
> **Q4**: In these lines, we suggest that there are more effective intermediate sequences than simply projecting each layer directly onto the final output. One possible approach is to leverage training results, as we have done in this paper. Alternatively, using prior knowledge about the network structure or specific application could help establish more refined bounds. Once this information of the structure of intermediate layers is provided, one can utilize the same scheme as YES-k to obtain YES-SIGMA bounds.
>
> **Q6**: We kindly refer the reviewer to our response in W7.
>
> **Q7**: The primary objective of a NN is to map inputs to $\mathbf{Y}$. A proven method to enhance this process is by deepening the network and introducing nonlinearity, which enables the model to capture complex features more efficiently. Alternatively, a straightforward but less optimal approach is to project each layer directly onto the output, disregarding the benefits of deeper network structures. Interestingly, our findings indicate that the YES-0 bound decreases as the network depth increases, even when each layer is independently projected onto the final output.

---

> ### Author Response · Authors · 2024-11-23
>
> **W10**: Thanks for the comments. You would likely agree that when the network is optimized, the optimal corresponding intermediate mapping points $\mathbf{Y}_k$ (the outcomes of the layers) are typically not equal to $\mathbf{Y} $. In fact the role of training is to find the optimal intermediate mapping points. I think you will also agree that, if we knew what intermediate mapping points $\mathbf{Y}_k$ are located at optimal training, we could have created much better bounds—intuitively, the closer the chosen $\mathbf{Y}_k$ to the optimal values, the better the bounds our YES framework can create. That is why we have a mechanism to import data from the training process as well, to enhance the bounds at the same time the training itself is getting better.

---

> > ### Comment · Reviewer_gBVS · 2024-11-25
> >
> > I thank the authors for their detailed clarifications.
> >
> > I continue to be skeptical of the papers contributions. Many of the clarifications are not able to address my concerns sufficiently. For instance, the cited passage from SB 1047 is irrelevant to model training diagnostics and I do not expect legislation on the training loss plateauing in a certain region.
> >
> > I would like to remind the authors that the appendix is still missing.
> >
> > My main criticisms vis-a-vis the contributions, soundness and strong claims made by the paper remain unaddressed. I will maintain my critical evaluation.

---

> > > ### Author Response · Authors · 2024-11-25
> > >
> > > Thank you for your continued feedback and for expressing your concerns about our paper. We appreciate the opportunity to clarify our contributions and address the points you've raised.
> > >
> > > We understand that our previous reference to SB 1047 may not have effectively illustrated the relevance of our work. Rather, we aimed to highlight the broader movement towards AI safety and the importance of developing technical tools that can support future regulatory efforts.
> > >
> > > Our Main Point: While there may not be specific legislation regulating training quality at this moment, it is crucial for the research community to proactively develop technical solutions that make such regulation feasible in the future. Effective regulation must refer to measurable benchmarks to assess compliance and performance. Our work contributes to this foundational groundwork by providing a practical and quantifiable method for monitoring training quality.
> > >
> > > Impact of Our Work: The YES Training Bounds framework offers a novel, data-aware method for real-time monitoring of neural network training quality. This benchmark can serve as a reference point for practitioners and potentially for future regulatory standards to ensure AI systems are trained properly. Our framework could assist in defining industry best practices and standards for training quality assessment. By detecting suboptimal training behaviors, such as loss plateaus in non-optimal regions, practitioners can intervene promptly to adjust training strategies. This reduces the risk of deploying AI systems that have been inadequately trained.
> > >
> > > Addressing Your Concerns Directly: While we do not anticipate legislation specifying technical details like "training loss plateauing," regulators will require measurable criteria to assess AI systems' safety and effectiveness. The YES bounds provide measurable criteria that regulators can refer to, offering a way to standardize training quality assessments across different models and applications.
> > >
> > > Conclusion: We believe that for reasons that include our initial phrasing of the response the impact of the work was underestimated. We hope that this clarification addresses your concerns and demonstrates the value and relevance of our contributions. We are committed to advancing the field in ways that are both technically meaningful and practically impactful.
> > >
> > > Thank you again for your thoughtful feedback.
> > >
> > > Sincerely,
> > >
> > > The Authors

---

> > > > ### Comment · Reviewer_gBVS · 2024-12-02
> > > >
> > > > I thank the authors for the further clarifications and appreciate the continued friendly and polite tone throughout the peer review process despite the extensive criticism.
> > > >
> > > > I maintain my view on the paper despite these clarifications. I suggest the reviewers provide a very detailed step-by-step reasoning for the YES bound.
> > > >
> > > > I want to give one example how this might look: Take a NN used as diagnostic tool for cancer based on images. Before deployment the model is examined in many ways: sensitivity and specificity of predictions, model performance under distribution shift, fairness criteria are studied, system-level risk for adversaries is assessed, assessment of uncertainty metrics is done to create a decision risk model (low training loss can btw be associated with worse uncertainty calibration), all of these are validated in a prospective study (e.g. randomized controlled double-blind clinical trial) establishing patient utility. What can the YES bound provide in terms of safety that weighs on the decision to approve the model for clinical use? This really just is one example, but the point is: what does this add that other methods cannot do. Why is the YES sub-optimality so important *beyond* other measurements?
> > > >
> > > > Once again, I thank the author vor engaging with the reviewers.

---

> > > > > ### Author Response · Authors · 2024-12-02
> > > > >
> > > > > Thank you for your valuable feedback. The primary focus of our paper is to monitor the training process's performance using a provided heuristic solution rather than assessing fairness or reliability during the testing phase. We aim to offer a straightforward, real-time method for evaluating training performance across epochs.
> > > > >
> > > > > Based on your comment, we want to clarify that we do not analyze a pre-trained neural network for the testing stage. Instead, we continuously monitor the training process. If the solution falls within the "green region" during training, we can state that it is suitable for use in the test phase.
> > > > >
> > > > > From our understanding of your concern, your focus is on the reliability and performance of the neural network during testing. Could you please clarify whether you recommend adding fairness and reliability assessments during the testing phase in addition to monitoring the training process?

---

### Official Review · Reviewer_jcCC · 2024-11-03

**Soundness:** 1
**Presentation:** 2
**Contribution:** 2
**Rating:** 3
**Confidence:** 2

**Summary:**

This paper introduces YES training bounds, a framework for real-time, data-aware certification and monitoring of neural network training. The framework evaluates data utilization efficiency and optimization dynamics, providing insights into training progress and detecting suboptimal behavior. The paper validates the YES bounds using synthetic and real data experiments, offering a tool for certifying training quality and guiding performance enhancements.

**Strengths:**

The proposed system's clarity is enhanced by the color-coded cloud-based monitoring system, which makes it intuitive for practitioners to interpret training status visually.

**Weaknesses:**

This paper has over-claimed its applicability, especially in model robustness and safety. None of the experiments discuss model safety and robustness. Accepting this paper unchallenged may send the wrong signal that the proposed method for enhancing model safety or robustness has been vetted, which it has not due to the omissions of related experiments.

**Questions:**

- Can you clarify how the YES training bounds directly contribute to improvements in model robustness and safety?

---

> ### Author Response · Authors · 2024-11-23
>
> We appreciate the reviewer’s insightful comments. We would like to highlight that model safety and robustness have multiple key elements, with training quality being a cornerstone that significantly influences reliability. If the training performance is poor, it is unreasonable to expect meaningful or reliable results from the model. In this paper, our primary objective is to monitor the quality of the training process. Leveraging our proposed mathematically grounded framework, we introduce the YES bounds and their associated cloud system, which serve as a sanity check for the optimizer. While we agree with the reviewer that we did not exhaustively show the impact of our bounds on test results, we have results (Fig 4 of Supplemental material) that show the correlation between training quality and test outcomes. We will further include MNIST test results for both training and test stages in the revised manuscript. We appreciate your comments bringing to our attention that without clarifying how central training quality is in model reliability, the contribution may be understood as over-stated. We hope that our clarifications have been helpful. Thank you!

---

> ### Comment · Reviewer_jcCC · 2024-12-03
>
> I have carefully read the authors' rebuttal as well as other reviewers' feedback. I recommend the author conduct more studies on how prior art addresses model safety, it is not enough to simply claim that training quality significantly affects reliability, and therefore think that the proposed method will benefit model reliability. The author should make significant revisions to the paper to better reflect the true contribution of the paper.

---

### Author Response · Authors · 2024-11-23

We sincerely appreciate the time and effort the review panel has dedicated to evaluating our submission, as well as the valuable feedback provided. Your insights have been instrumental in refining our work.

In the race to make AI systems more reliable, the focus has predominantly been on generalization. However, we pose an essential question: how can we ensure reliable AI when the training process itself lacks rigorous evaluation and guarantees? This gap is significant and demands a solution. Our work directly addresses this challenge by introducing a novel, practical framework that enhances the reliability of the training process—an often-overlooked but critical aspect of building safe and robust AI systems.

Specifically, our contributions include:
1. YES training bounds – A real-time, data-aware framework to evaluate how well training is progressing.
2. A color-coded cloud system – Visualize in real-time whether your training is effective (green), needs caution (sub-optimal, yellow), or is outright poor (red).

These contributions go beyond academic curiosity; they represent a practical and impactful solution to a crucial problem. By leveraging our framework, practitioners can:
- Monitor training issues as they happen.
- Gain actionable insights into training quality.
- Defend the training received against regulatory or industry benchmarks for safety and performance.

We have worked to address the reviewer comments and suggestions below and hope that our responses are satisfactory. We are confident that our framework offers an indispensable addition to the AI community, one that meets both the technical rigor and practical utility expected of ICLR contributions.

---

### Note · Authors · 2025-01-28

I have read and agree with the venue's withdrawal policy on behalf of myself and my co-authors.